# In situ acid etching boosts mercury accommodation capacities of transition metal sulfides

Hailong Li [1], Jiaoqin Zheng[1], Wei Zheng[1], Hongxiao Zu[1], Hongmei Chen[1], Jianping Yang[1], Wenqi Qu[1], Lijian Leng[1], Yong Feng[2] & Zequn Yang[1] ✉

Transition Metal sulfides (TMSs) are effective sorbents for entrapment of highly polluting thiophiles such as elemental mercury ($Hg^0$). However, the application of these sorbents for mercury removal is stymied by their low accommodation capacities. Among the transition metal sulfides, only CuS has demonstrated industrially relevant accommodation capacity. The rest of the transition metal sulfides have 100-fold lower capacities than CuS. In this work, we overcome these limitations and develop a simple and scalable process to enhance $Hg^0$ accommodation capacities of TMSs. We achieve this by introducing structural motifs in TMSs by in situ etching. We demonstrate that in situ acid etching produces TMSs with defective surface and pore structure. These structural motifs promote $Hg^0$ surface adsorption and diffusion across the entire TMSs architecture. The process is highly versatile and the in situ etched transition metal sulfides show over 100-fold enhancement in their $Hg^0$ accommodation capacities. The generality and the scalability of the process provides a framework to develop TMSs for a broad range of applications.

Transition metal sulfides (TMSs) are materials with the generalized formula MSx, where M represents the metal atoms from groups four to ten, and $X$ denotes the stoichiometric number of sulfides ($X$ ranges from 1/2 to 2 in most cases)[1]. Compared to the oxide, selenide, and telluride counterparts of transition metal cations and the group-fourteen anions, most TMSs attainable under mild synthetic conditions allow for more process optimization for use in energy conversion and environmental remediation. In TMSs, the surface metals and sulfide terminals being positively and negatively charged generally feature depletion and accumulation of electrons and accommodate adsorbates with the opposite electronic properties and promote their subsequent transformations[2]. The foundational adsorption process and its characteristics, including but not limited to the coordination pattern, adsorption energy, and electron distribution, further permit or inhibit the use of TMSs in a range of processes, e.g., as catalysts. Elucidating the adsorption of heterogeneous components on TMSs is thus required for understanding the subsequent conversions of the guest components.

With their enriched contents of surface sulfides, TMSs are effective sorbents with which to trap thiophilic counterparts and initiate transformations[3]. Taking this inspiration, the adsorption of elemental mercury ($Hg^0$), both a thiophilic element and a global pollutant[4,5], on TMSs has been examined extensively to enable oxidation and stabilization, mimicking the natural processes of mercury deposition[6]. Since 2016, when TMSs were proposed to be promising candidates for $Hg^0$ capture, various TMSs, including zinc sulfide (ZnS)[7], copper sulfide (CuS)[8], cobalt sulfides ($CoS_x$)[9], iron sulfides ($FeS_x$)[10], molybdenum disulfide ($MoS_2$)[11], etc.[12], have been scrutinized to determine their $Hg^0$ adsorption capacities by members of the environmental chemistry research community. In the post-Minamata Convention era, TMS-based techniques are regarded as promising alternatives to the use of activated carbons because more than 130 worldwide signatories of the Convention scheduled to come into effect in 2017 simultaneously emphasized the need for efficient abatement of mercury pollution and centralized recycling of mercury resources[13,14]. These requirements

[1]School of Energy Science and Engineering, Central South University, Changsha 410083, China. [2]Environmental Research Institute, South China Normal University, Guangzhou 510631, China. ✉e-mail: Zequn_Yang@hotmail.com

have prioritized the use of TMSs in $Hg^0$ capture because (1) TMSs hold great potential to accommodate much more mercury than activated carbons due to the abundance of ligands with affinity towards Hg in the TMSs, and (2) TMSs that immobilize a high volume of mercury are ideal sources for stable mercury storage and efficient mercury recycling through smelting[7,8].

Although TMSs have been extensively screened for use in $Hg^0$ adsorption, convincing justifications for the use of TMSs other than CuS, if any, remain to be developed considering their moderate $Hg^0$ capture capacities. The environmental chemistry community has long recognized that CuS has a high $Hg^0$ accommodation capacity (approximately 0.1 gram of $Hg^0$ per gram of CuS), ensuring the feasibility of using TMSs in practical $Hg^0$ removal and recovery scenarios[15]; however, this value far exceeds the current capabilities of most other TMSs (as shown in Supplementary Table 1). Typical earth-abundant TMSs such as ZnS even exhibit $Hg^0$ adsorption capacities of approximately 0.001 gram of $Hg^0$ per gram of TMS[16,17], two orders of magnitude inferior to that of CuS and merely comparable to those of activated carbons. The limited mercury accommodation capacities of these TMSs makes applications impractical because (1) the syntheses of TMS-based sorbents are thought to be more complex than those of activated carbons, and (2) mercury recycling from TMSs containing low volumes of mercury is costly and impractical. The status quo provides a predicament, since CuS commercialization has come closer with no alternative TMS as a backup, which compromises the cost-effective principle that one must consider the range of geographical abundances and localized preferences of TMSs.

Systemizing the various TMSs that exhibit $Hg^0$ capture capacities comparable to that of CuS into a range of practical solutions, in spite of their different requirements and scenarios, is one of the most challenging tasks facing the mercury chemistry community because there are intrinsic properties of CuS that are not found in other TMSs; these include the layered structure and well-defined boundary inhibiting over-agglomeration and abundance of undercoordinated sulfur atoms serving as effective electron acceptors[18]. Although valuable lessons have been taken from the CuS case, i.e., the structural motifs with hierarchical benefits that enable both transportation and oxidation of $Hg^0$, which may be critical to improving the $Hg^0$ capture capabilities of sulfides other than CuS, there is no rational method for achieving generalized syntheses of TMSs to this end. Adjustments such as amorphization, heteroatom doping, support addition, and oxygen incorporation have been used extensively to enhance the performance of TMSs, but these TMSs have rarely exhibited $Hg^0$ accommodation capacities approaching the threshold of $0.1\,g\,g^{-1}$ [10,16,19–23]. This fundamental limitation suggests the need for a synthetic method that provides an abundance of the advantageous structural motifs in TMSs.

In this work, a scalable in situ acid-etching method was developed to construct structural motifs in TMSs by recognizing that most earth-abundant TMSs are soluble in dilute acid (with $K_{sp} > 10^{-30}$). A favorable environment with mass transfer channels and structural defects promoting electron exchange was formed by thermodynamically spontaneous dissolution of the TMSs, and optimization was achieved by adjusting the reaction conditions to control the etching kinetics. Compared to post-etched TMSs formed via an ex situ method, i.e., mixing acids with the TMS instead of its precursors (as shown in Supplementary Fig. 1), in situ etching worked throughout the sample preparation process and thus introduced structural motifs with higher homogeneities and abundances into the samples. As the capacities of TMSs for adsorption (pre-adsorption) of different atoms/molecules other than $Hg^0$ typically require improvements in otherwise identical manipulations, the generality of this method in promoting the adsorption capabilities of TMSs may attract broad interest among the environmental chemistry community and enable flexible use for adsorption of heterocomponents other than $Hg^0$.

## Results

Considering the earth abundance of ZnS, it was taken as a typical example with which to study the fundamentals of the in situ acid etching method, and extensive applicability of this method was confirmed after the critical parameters were optimized and the mechanisms involved were elucidated. Pristine ZnS was obtained through a precipitation method, which is well known as a scalable method that is preferred in industry, by mixing stoichiometric solutions of zinc sulfate ($ZnSO_4$) and sodium sulfide ($Na_2S$). The X-ray diffraction pattern (XRD) for the pristine ZnS was indexed to the cubic phase of ZnS exhibiting the space group F-43 m (PDF #03-065-0309), and only the (100), (220), and (311) crystal surfaces were detected (as shown in Supplementary Fig. 2)[24]. To conduct in situ acid etching by mixing the $ZnSO_4$ and $Na_2S$ solutions, different amounts of sulfuric acid ($H_2SO_4$) were added to the former solution. The ZnS samples obtained in the presence of 0.0184, 0.092, 0.184, and $0.368\,mol\,L^{-1}$ $H_2SO_4$ were denoted as 0.0184-ZnS, 0.092-ZnS, 0.184-ZnS, and 0.368-ZnS, respectively. In addition, as a comparison, an ex situ acid etching method was used to treat the pristine ZnS, i.e., $0.368\,mol\,L^{-1}$ $H_2SO_4$ was mixed with pristine ZnS instead of its precursors, and the sample obtained was denoted as ex0.368-ZnS. As shown in Supplementary Fig. 2, 0.0184-ZnS, 0.092-ZnS, 0.184-ZnS, 0.368-ZnS, and ex0.368-ZnS shared identical peaks with locations comparable with those of pristine ZnS, indicating a generally unchanged crystalline phase was formed during the etching process.

The $Hg^0$ adsorption behaviors of pristine ZnS, 0.0184-ZnS, 0.092-ZnS, 0.184-ZnS, 0.368-ZnS, and ex0.368-ZnS were evaluated preliminarily with a fixed-bed reaction system (as shown in Supplementary Fig. 3) to quantify the effects of acid concentration. As shown in Supplementary Fig. 4, when pristine ZnS was employed, the normalized $Hg^0$ outlet concentration was maintained at approximately 0.8, specifying a $Hg^0$ capture efficiency of 17.3% within 120 min. With the in situ addition of only $0.0184\,mol\,L^{-1}$ $H_2SO_4$, the normalized $Hg^0$ outlet concentration was dramatically decreased to approximately 0.2, and the $Hg^0$ adsorption efficiency was significantly increased to 79.7% when 0.0184-ZnS was adopted, indicating the critical role of in situ acid etching in improving the $Hg^0$ adsorption capacities of the metal sulfides. Increasing the dosages of $H_2SO_4$ to 0.092, 0.184, and $0.368\,mol\,L^{-1}$ generated stepwise enhancements in the $Hg^0$ capture efficiencies of the ZnS, which finally reached 100% within 2 h when the concentration of $H_2SO_4$ was $0.368\,mol\,L^{-1}$. Although this trend suggested that $H_2SO_4$ concentrations exceeding $0.368\,mol\,L^{-1}$ might cause further performance improvements in ZnS used for $Hg^0$ adsorption, the loss of ZnS caused by the excess $H_2SO_4$ nevertheless made further increases in $H_2SO_4$ concentration impractical. When the dosages of $H_2SO_4$ were 0, 0.092, 0.184, and $0.368\,mol\,L^{-1}$, the conversion rates for the precursors equaled 95%, 90%, 70%, and 50%, respectively. It was found that in $0.368\,mol\,L^{-1}$ acid, the precursor conversion rate were decreased by approximately one-fold compared to that for pristine ZnS because of dissolution of the ZnS and evaporation of the hydrogen sulfide ($H_2S$) caused by the acid (which is briefly illustrated by the following reactions).

$$ZnS + H^+ \rightarrow HS^- + Zn^{2+} \qquad (1)$$

$$HS^- + H^+ \rightarrow H_2S(g) \qquad (2)$$

$$ZnS + 2H^+ \rightarrow Zn^{2+} + H_2S(g) \qquad (3)$$

It should also be noted that when the $H_2SO_4$ concentration was increased to $-0.75\,mol\,L^{-1}$, negligible ZnS was produced by the synthetic process. The weight loss occurring during etching indicated that the $Hg^0$ adsorption performance and the yield of ZnS should be balanced in practical scenarios. The impracticality of increasing the

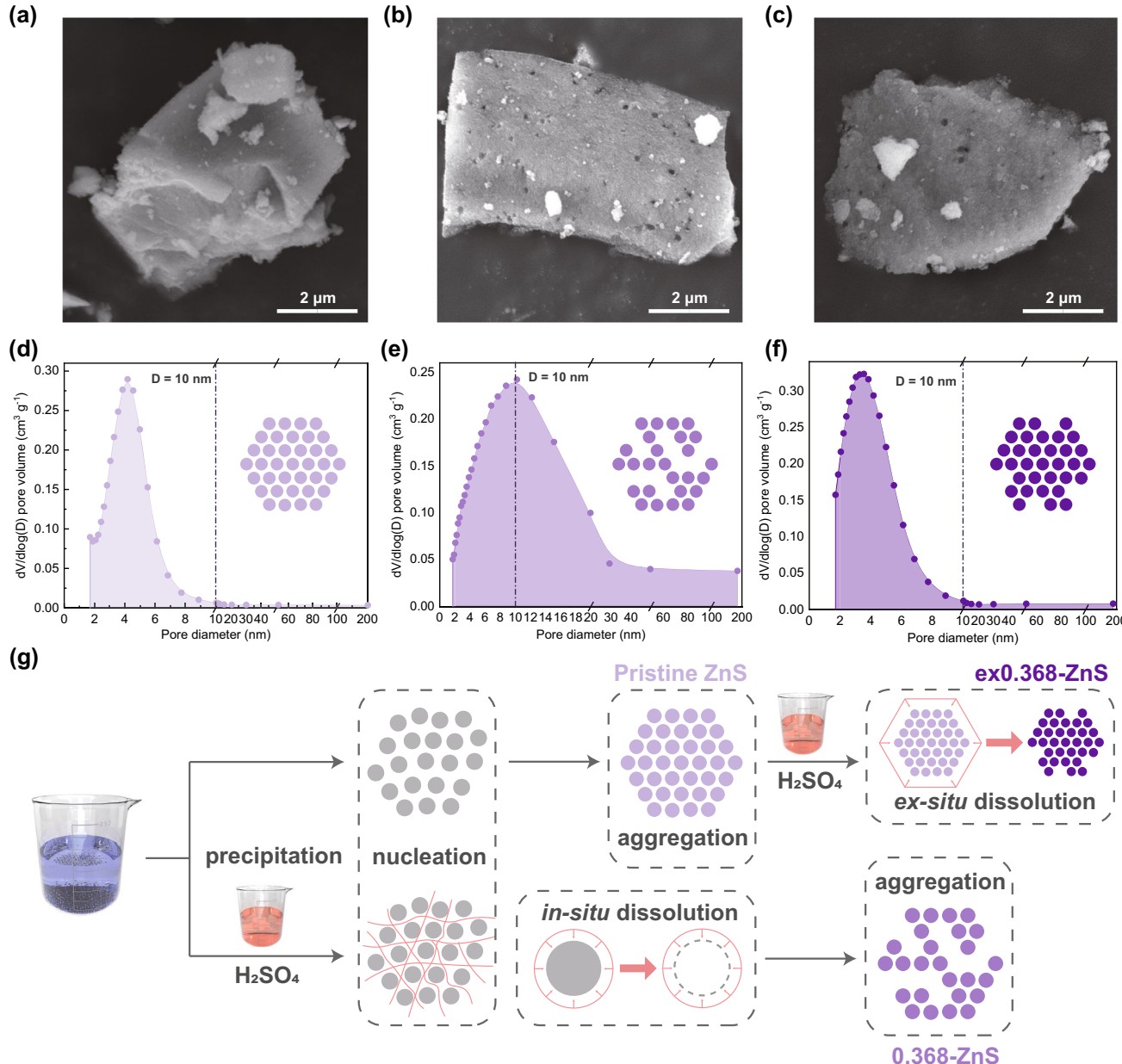

**Fig. 1 | Textural properties of different ZnS samples.** SEM images of (**a**) pristine ZnS, (**b**) 0.368-ZnS, and (**c**) ex0.368-ZnS; pore distributions in (**d**) pristine ZnS, (**e**) 0.368-ZnS, and (**f**) ex0.368-ZnS; and (**g**) diagrammatical illustration of the formation of varied mass transfer environments in pristine ZnS, 0.368-ZnS, and ex0.368-ZnS.

acid concentration to a level beyond 0.368 mol L$^{-1}$ made 0.368-ZnS, which fully manifested the advantages of in situ etching, the preferred sample for conducting a mechanistic exploration.

The scanning electron microscope (SEM) images of pristine ZnS, 0.368-ZnS, and ex0.368-ZnS (as shown in Fig. 1(a−c)) confirmed that the surfaces of the agglomerated 0.368-ZnS and ex0.368-ZnS nanoparticles were porous in nature, whereas the surface of the pristine ZnS was more condensed. Comparisons of the BET surface areas, pore volumes, and average pore diameters among pristine ZnS, 0.368-ZnS, and ex0.368-ZnS are shown in Supplementary Fig. 5. Pristine ZnS, 0.368-ZnS, and ex0.368-ZnS exhibited BET surface areas of 161.6, 112.5, and 194.3 m$^2$ g$^{-1}$, pore volumes of 0.11, 0.23, and 0.17 cm$^3$ g$^{-1}$, and average pore diameters of 3.21, 6.73, and 3.31 nm, respectively. Compared with those of pristine ZnS and ex0.368-ZnS, the Brunauer–Emmett–Teller (BET) surface area of 0.368-ZnS was slightly lower, while its pore volume and average pore diameter were significantly

higher. This trend indicated that the decreased BET surface area of 0.368-ZnS might be attributed to characteristic changes in the pores. The N$_2$ adsorption-desorption patterns for pristine ZnS, 0.368-ZnS, and ex0.368-ZnS further supported this notion. As shown in Supplementary Fig. 7, pristine ZnS and ex0.368-ZnS exhibited identical type IV isotherms in the curves for N$_2$ adsorption-desorption with remarkable hysteresis loops at P/P$_0$ = 0.4–0.7, suggesting enrichment of the mesopores[25]. The in situ etching process nevertheless provided a sample (0.368-ZnS) exhibiting a type IIIB isotherm curve with a hysteresis loop at P/P$_0$ at 0.5–0.9, which was indexed to show a combination of mesopores and macropores[26].

The meso/macropore distribution patterns of pristine ZnS, 0.368-ZnS, and ex0.368-ZnS are shown in Fig. 1(d−f) and further support the textural changes occurring in ZnS with the different etching scenarios. As shown, pore diameters ranging from 2 to 10 nm dominated in pristine ZnS, and the peak value was located at approximately 4.5 nm.

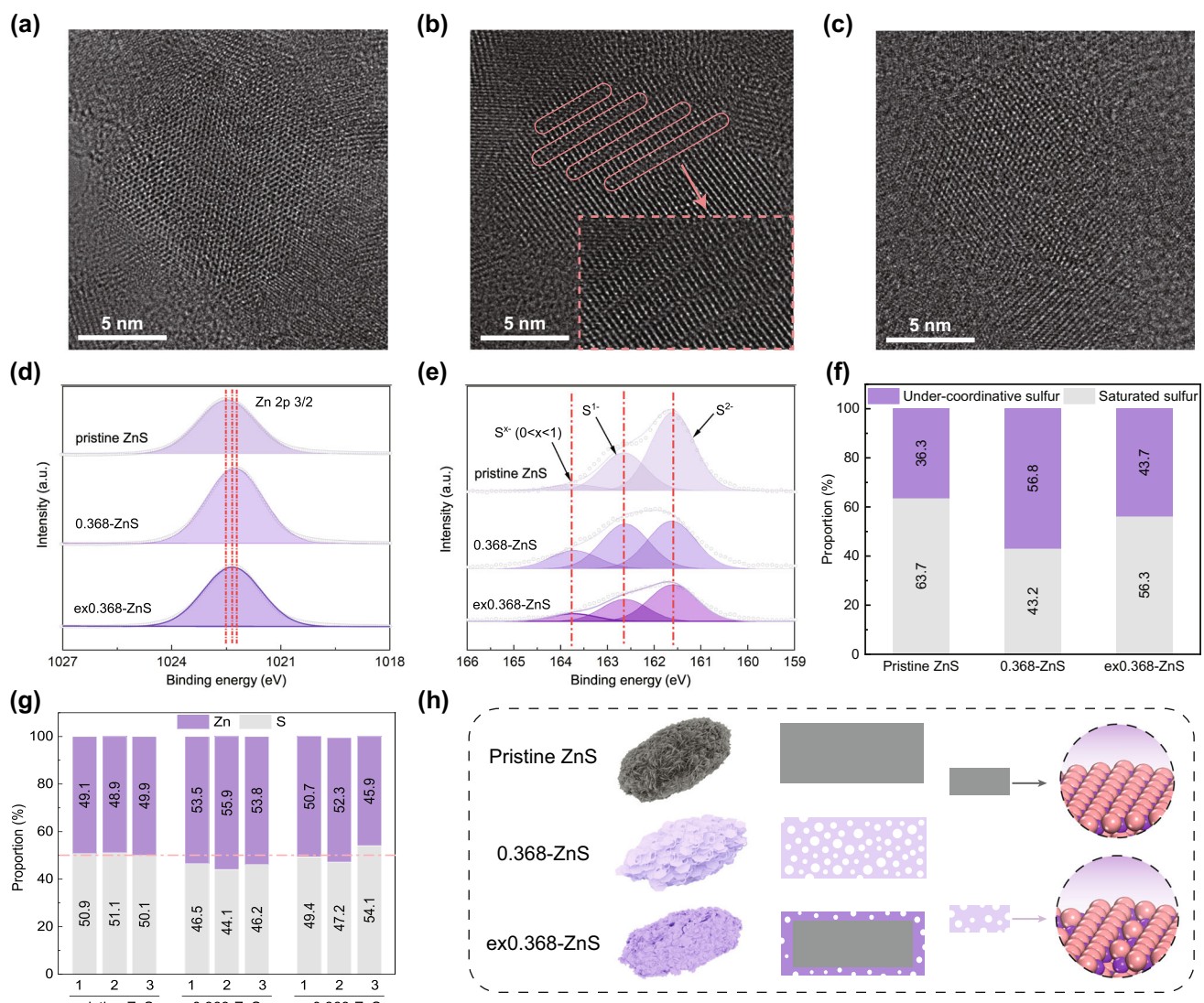

**Fig. 2 | Compositional properties of different ZnS samples.** STEM-HAADF images of pristine (**a**) ZnS, (b) 0.368-ZnS, and (**c**) ex0.368-ZnS (the highlighted regions in panel (**b**) represent the defect-rich area); XPS data showing the (**d**) Zn *2p* and (**e**) S *2p* binding energies of different samples; the compositional proportions of (**f**) various sulfur species and (**g**) Zn/S in different samples; and (**h**) diagrammatical illustration of the structural and atomic defects in different samples.

After treatment via ex situ etching, the pore distribution in ex0.368-ZnS showed negligible differences compared to that of pristine ZnS. However, in situ etching dramatically changed the pore distribution of ZnS, and pores with diameters of 2–200 nm resulted in 0.368-ZnS with a peak value at approximately 10 nm. The textural difference indicated that only in situ etching enriched the structure of ZnS with hierarchical pores. Although the SEM images showed that the agglomerated 0.368-ZnS and ex0.368-ZnS nanoparticles shared a similar external appearance, their internal structures and homogeneities might nevertheless vary significantly.

A probable interpretation of this variety is shown in Fig. 1(g). During the synthesis of 0.368-ZnS, the nucleation process was accompanied by the in situ dissolution of ZnS, which was illustrated by Eqs. (1)-(3). The in situ dissolution of ZnS instantly created voids separating the ZnS nanoparticles that were subsequently formed. In this case, as the dissolution of ZnS nanoparticles occurred constantly during nucleation, the undissolved ZnS nanoparticles were efficiently and spatially isolated by the voids, which thus ensured homogenous formation of both structural and surface pores. This favorable homogeneous texture aided the diffusion of heterocomponents into 0.368-ZnS, hence improving the Hg$^0$ adsorption performance. However,

ex0.368-ZnS was well-crystallized when immersed in acids. In this case, the acid could not efficiently penetrate into the interior parts of the nanoparticles, thus leading to an inhomogeneous distribution in the porous architecture, i.e., only the surfaces of the agglomerated nanoparticles were etched, and no significant changes were observed in their textural properties. The unchanged interior structure containing limited pores might block the transportation of Hg$^0$ in ex0.368-ZnS, hence compromising its Hg$^0$ capture performance.

To obtain a nanoparticle-level understanding of the influence of acid etching, transition electron microscopy (TEM) and high-resolution TEM (HRTEM) images were collected (as shown in Supplementary Fig. 7 and Fig. 2(a–c)). The ZnS particles were irregularly shaped with an average diameter in the sub-10 nm range. The well-dispersed 0.368-ZnS nanoparticles suspended in alcohol additionally indicated the lower density structure of 0.368-ZnS compared to those of pristine ZnS and ex0.368-ZnS. The lattice fringes of pristine ZnS (0.31 nm), 0.368-ZnS (0.32 nm), and ex0.368-ZnS (0.31 nm) shown in the HRTEM images of a single nanoparticle corresponded to the (111) crystalline planes of cubic-phase ZnS[27]. Notably, unlike pristine ZnS and ex0.368-ZnS, which were relatively intact, the nanoparticles of 0.368-ZnS were defect rich (as highlighted in Fig. 2(b)). The structural

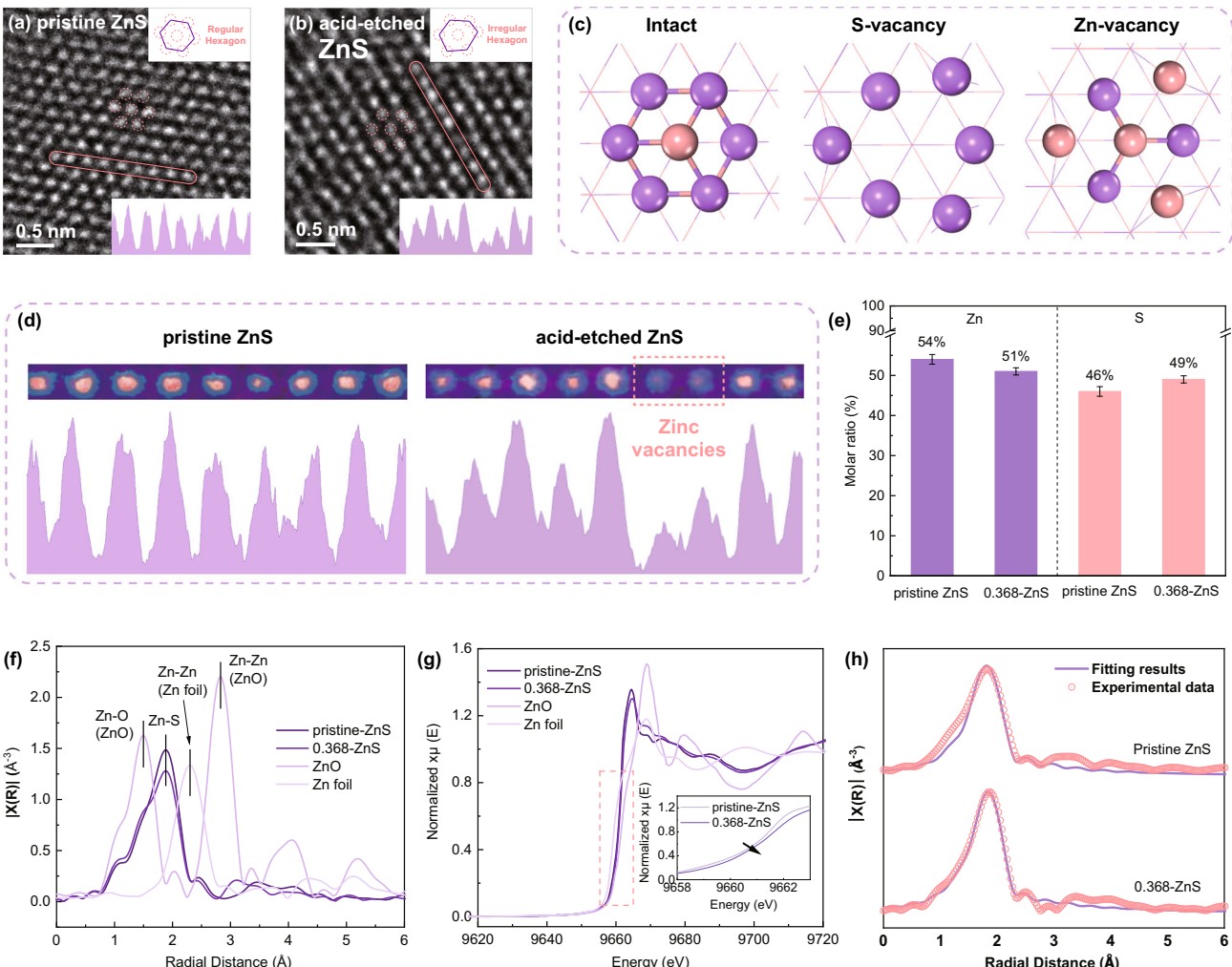

**Fig. 3 | Defect-rich property of acid-etched ZnS. a, b** STEM-HAADF images of pristine and acid-etched ZnS (the highlighted regions denote the regions used for atomic mapping); (**c**) geometrical optimization results for intact, S-vacancy, and Zn-vacancy surfaces (pink: S, violet: Zn); (**d**) simulated elemental maps of Zn atoms in pristine and acid-etched ZnS; (**e**) compositional ratios of Zn and S atoms in pristine and acid-etched ZnS, as obtained by ICP–MS; (**f**) EXAFS patterns for pristine and acid-etched ZnS; (**g**) Zn K-edge XANES patterns for pristine and acid-etched ZnS; and (**h**) the corresponding EXAFS R space fitting results for pristine and acid-etched ZnS. The error bars in panel (**e**) correspond to the standard deviation of three independent measurements, and the data are presented as mean values.

voids created more undercoordinated edge sites that enabled electron transport between the sorbents and adsorbates[28–30]. X-ray photoelectron spectroscopy (XPS) showed the undercoordinated zinc and sulfur atoms in 0.368-ZnS (as shown in Fig. 2(d) & (e) & Supplementary Fig. 8). The Zn $2p_{3/2}$ XPS peaks for pristine ZnS, 0.368-ZnS, and ex0.368-ZnS were located at 1022.48, 1022.26, and 1022.34 eV, respectively, indicating that the coordination numbers of Zn in the different ZnS samples decreased in the order pristine ZnS > ex0.368-ZnS > 0.368-ZnS[31]. In addition, the S $2p$ peaks for 0.368-ZnS also suggested that the number of undercoordinated sulfurs was dramatically increased after treatment by in situ etching[8]. Notably, the proportion of undercoordinated sulfurs on the surface was 36.3% for pristine ZnS and was 56.8% for 0.368-ZnS (as shown in Fig. 2(f)). No signal centered at ~168 eV for $S^{6+}$ was observed,[15] which excluded interference from $ZnSO_4$ during the $Hg^0$ removal process.

As the presence of structural voids is generally accompanied by native atom vacancies because point defects are fundamental for the formation of volume voids[32], it is reasonable to speculate that there were also numerous atomic vacancies in 0.368-ZnS. Considering that the typical volume defects seen in ZnS keep the stoichiometries of zinc and sulfur unchanged[33], EDS point scanning was used to search for the formation of atomic defects (as shown in Supplementary Figs. 9-10).

The results shown in Fig. 2(g) confirmed the nonstoichiometric ratio of zinc and sulfur in 0.368-ZnS, while those in pristine ZnS and ex0.368-ZnS were maintained at 1:1. Furthermore, the EDS mapping results indicated the homogeneous Zn-enriched nature of 0.368-ZnS (as shown in Supplementary Fig. 11). The unusual overaccumulation of Zn in 0.368-ZnS implied the formation of S atom vacancies since $H_2S$ might have evaporated during nucleation of the ZnS (as indicated by reaction (1–3)). Thus, in addition to pore engineering, in situ etching might have also introduced defects (both structural and atomic defects) into 0.368-ZnS (as shown in Fig. 2(h)), hence increasing the number of undercoordinated sites that accelerated electron transport between the sorbents and adsorbates.

Since the EDS results only provided indicative evidence for the presence of undercoordinated sites in 0.368-ZnS and the species of the undercoordinated sites were unidentified, more atomic analyses were conducted (as shown in Fig. 3). The STEM-HAADF (scanning-transmission electron microscopy equipped with a high-angle annular dark-field detector,) images shown in Fig. 3(a) and (b) showed the configurations of Zn atoms on the surfaces of the pristine and acid-etched ZnS because the S atoms were generally invisible in STEM-HAADF images due to their low contrast. As shown, the Zn atoms in acid-etched ZnS exhibited two obvious changes compared to the pristine

sample, i.e., (1) the regular hexagonal arrangements of Zn atoms on the ZnS (111) surface became irregular, and (2) the contrast levels of several Zn atoms decreased. The first observation suggested an enrichment in S defects because only the presence of S defects in ZnS caused the displacement of Zn atoms (as shown by the DFT calculation results in Fig. 3(c)), while the second observation was much more straightforward in proving the presence of Zn vacancies. The simulated elemental maps based on the STEM-HAADF patterns of pristine and acid-etched ZnS clearly showed dramatically decreasing contrasts for the Zn atoms in acid-etched ZnS, which is probably attributable to the loss of surface Zn atoms (as shown in Fig. 3(d)). In this case, the STEM-HAADF image only showed Zn atoms located in the sublayers, which thus made the intensities of the Zn signals decrease. In addition, the inductively coupled plasma–mass spectrometry (ICP–MS) results also proved the nonstoichiometric nature of Zn and S atoms in the acid-etched ZnS (as shown in Fig. 3(e)). Specifically, in acid-etched ZnS, the S content was higher by 3% compared to that of the pristine ZnS, and this was accompanied by a corresponding decrease in the Zn content. The excellent agreement between the STEM-HAADF and ICP–MS results further demonstrated the Zn-defect nature of the acid-etched ZnS.

In addition, as mentioned above, the STEM-HAADF pattern for acid-etched ZnS indicated, qualitatively, the presence of S defects. To confirm this, the X-ray absorption spectra (XAS) of the Zn atoms in the pristine and acid-etched ZnS are shown in Fig. 3(f–h) to characterize their coordination environments. As shown in Fig. 3(f), the extended X-ray absorption fine structure (EXAFS) data showing the coordination environments around the Zn atoms indicated Zn-S coordination with radial distances ranging between those of Zn-Zn and Zn-O bonds. From the X-ray absorption fine structure spectroscopy (XANES) data for the Zn K-edge (as shown in Fig. 3(g)), the valences of the Zn atoms in both the pristine and acid-etched ZnS samples were higher than that seen in Zn foil but lower than that seen in ZnO, further confirming the formation of Zn-S bonds. In addition, it should be specially noted that the subfigure in Fig. 3(g) highlights the XANES patterns for pristine and acid-etched ZnS, and the valences of the Zn atoms in acid-etched ZnS were lower than those in the pristine samples, which indicated the formation of S defects that decreased the coordination numbers of Zn atoms in the acid-etched ZnS. The corresponding EXAFS R space fitting results quantified the coordination numbers of Zn atoms as 3.9 and 3.6 in pristine and acid-etched ZnS, respectively (as shown in Fig. 3(h)). Based on the excellent agreement between the EDS and XAS patterns, it is reasonable to conclude that in situ acid etching introduced S defects into the ZnS. Favorable structural motifs comprised of externally introduced pores and defects were thus constructed in 0.368-ZnS to enable adsorption with abundant migration pathways and active sites.

Since the adsorption capacities of functional materials are generally affected by two primary influential factors in practical application, i.e., diffusion and immobilization, they were considered separately by adopting different reaction systems to elucidate the mechanisms resulting in the performance enhancements. The nested-tube reaction system containing highly concentrated vapor-phase $Hg^0$ was suitable to promote the adsorption towards rate-irrelevant, and was thus used to characterize the abundance of active sites in the sorbents and exclude the influence of mass transfer (as shown in Supplementary Fig. 12)[24,34,35]. As shown in Supplementary Fig. 13, mercury temperature programmed desorption/decomposition (Hg-TPD) tests were conducted to quantify the amounts of mercury adsorbed by pristine ZnS, 0.368-ZnS, and ex0.368-ZnS pretreated in a nested-tube reactor. Compared to the pristine ZnS that adsorbed 215.1 μg of $Hg^0$, the amounts of $Hg^0$ adsorbed on 0.368-ZnS and ex0.368-ZnS were approximately eight- and threefold higher (1772.4 and 664.1 μg), indicating that, even though the influence of mass transfer was excluded and the porous structure did not work, the etched ZnS outperformed the pristine ZnS in $Hg^0$ capture. This

observation was generally in line with the XPS comparison between Hg-laden pristine ZnS and Hg-laden acid-etched ZnS (as shown in Supplementary Fig. 14(a)), which showed that the Hg signal in Hg-laden pristine ZnS was much weaker than that in Hg-laden acid-etched ZnS. The inherent reasons leading to this were attributed to the increased abundance of active sites in the acid-etched ZnS compared to the pristine ZnS, since that the influence of diffusion was generally excluded in this case. Although the specific active sites remain to be identified, the Zn 2p and S 2p XPS data for Hg-laden 0.368-ZnS indicated that mercury interacted strongly with the ZnS (as shown in Supplementary Fig. 14(b) & (c)).

From the characterization and performance results, it is obvious that ex0.368-ZnS combined the properties of both pristine ZnS and 0.368-ZnS, i.e., the external parts of the nanoparticles were etched by acid, while the internal parts maintained the characteristics of pristine ZnS. Thus, in the mechanistic discussions, only pristine ZnS and 0.368-ZnS were used to interpret why acid etching enriched the active sites. A slab model of the ZnS(111) surface was constructed based on the HRTEM results and the exposure ratios of different crystal surfaces were calculated with the classic BFDH model (a model suggested by Bravasis, Freidel, Donnary, and Harker), which generally supported an inverse correlation between the lattice fringes among the <hkl> planes and the exposure ratios of the corresponding facets (as shown in Supplementary Figs. 15 & 16). The theoretical exposure ratio of the ZnS (111) surface as obtained from the BFDH model was implemented in the Morphology Tools in Materials Studio 2020 and reached as high as ~79%. Both cationic and anionic defects were considered to approximate the experimental scenarios, and one of the surface Zn or S atoms was removed (as shown in Supplementary Fig. 17).

As shown in Supplementary Fig. 18, four possible $Hg^0$ adsorption sites were considered for the intact ZnS(111) surfaces (namely, hollow, Zn-top, S-top, and bridge sites), while two extra adsorption sites were included for the S-defect and Zn-defect Zn (111) surfaces (namely, S-defect and undercoordinated Zn-top sites or Zn-defect and undercoordinated S-top sites), and the adsorption configurations with the highest adsorption energies were taken as the final adsorption patterns. The electron density differences for intact, S-defect, and Zn-defect ZnS (111) surfaces were first analyzed to predict their affinities for $Hg^0$. As shown in Fig. 4(a), electrons were accumulated around the surface-exposed S atoms on the intact ZnS(111) surfaces. In contrast, on the S-defect and Zn-defect Zn (111) surfaces, electron traps were formed around the defect sites due to undercoordination. During the adsorption of $Hg^0$, electrons were donated to sorbent to effect oxidation of Hg; thus, the accumulation of electrons generally impeded this electron donation process. As expected, the most preferred configurations for $Hg^0$ adsorption on the intact, S-defect, and Zn-defect surfaces exhibited adsorption energies of 102.3, 149.2, and 355.1 kJ mol$^{-1}$ (as shown in Fig. 4(b)), which were in agreement with the predictions from the electron density differences. The Hirshfeld charges for adsorbed Hg on the intact, S-defect, and Zn-defect surfaces were 0.29, 0.35, and 0.49 e, respectively, indicating that the electron-deficient surface attracted more electrons from $Hg^0$, hence making the Hg more positively charged. The partial density of state (PDOS) patterns for the Hg atoms and the adjacent Zn or S atoms are shown in Supplementary Fig. 19. The PDOS analysis indicated that the interaction intensities between the electrons of the Hg atoms and the electrons of the adjacent Zn or S atoms also decreased in the order Zn-defect > S-defect > intact, which confirmed the important role that undercoordinated sites played in $Hg^0$ adsorption.

As is well known, comparing the $Hg^0$ adsorption energies among different sorbents is relatively meaningless because the adsorption capacities of various sorbents are not necessarily proportional to their affinities for $Hg^0$. However, the affinity between the sorbent surface and $Hg^0$ may influence the possibility of achieving multilayer adsorption, hence changing the adsorption capacity. As shown in Fig. 4(a), the

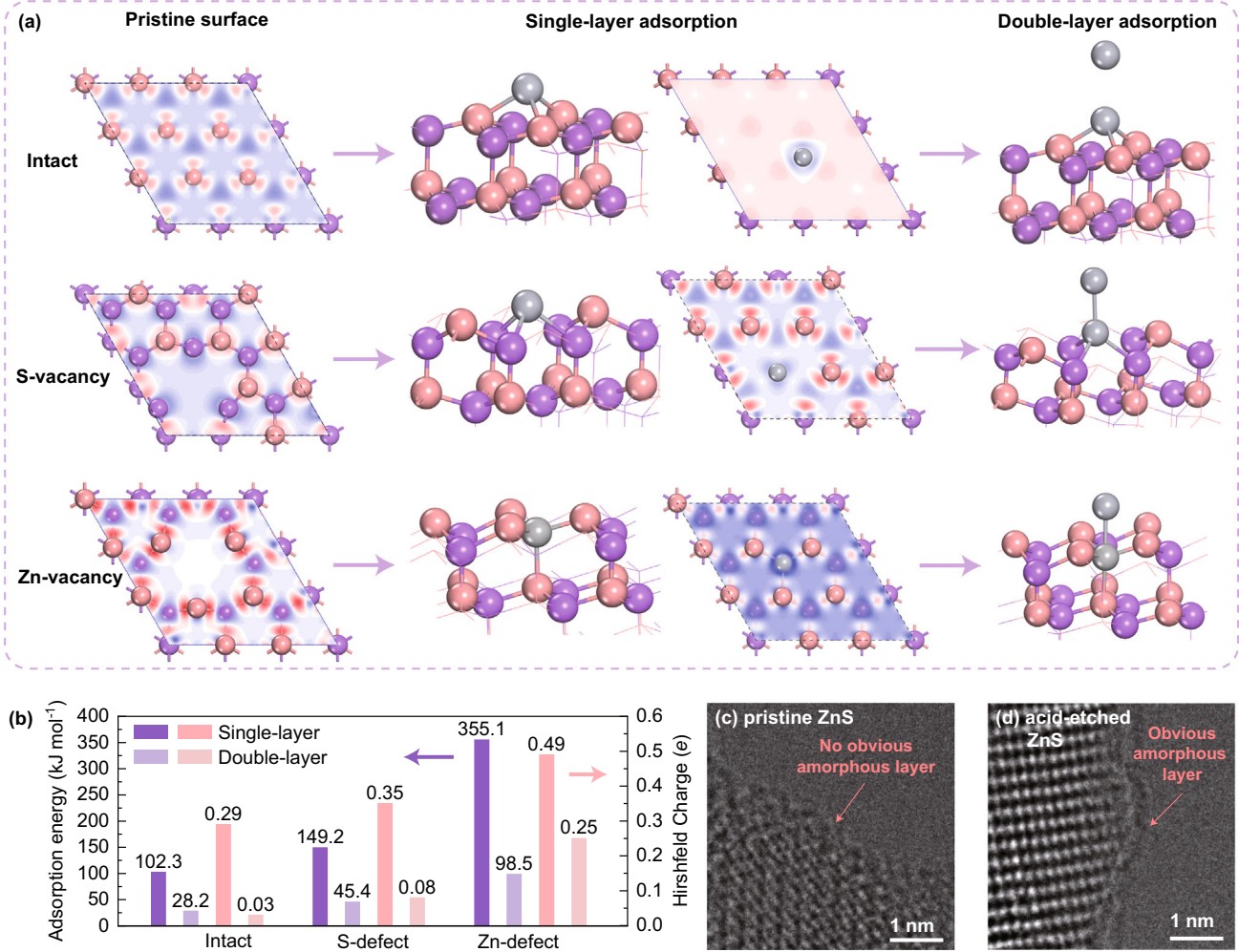

**Fig. 4 | Hg⁰ adsorption behaviors over different ZnS samples. a** Electron density differences (red: electron-rich, blue: electron-deficient) for intact, S-vacancy, and Zn-vacancy surfaces and the Hg adsorption patterns of intact, S-vacancy, and Zn- vacancy surfaces; (**b**) adsorption energies and atomic charges of Hg on intact, S-vacancy, and Zn-vacancy surfaces; and (**c**) & (**d**) STEM-HAADF images of pristine and acid-etched ZnS (Hg-laden samples).

electron density differences were also analyzed for intact, S-defect, and Zn-defect ZnS(111) surfaces with one Hg atom adsorbed. Electron-deficient areas formed around the Hg atoms adsorbed on the intact, S-defect, and Zn-defect ZnS(111) surfaces. As with the Hirshfeld charge results, the deficiencies for electron densities around the Hg atoms adsorbed on different surfaces decreased in the order Zn-defect > S-defect > intact, which was attributed to the different abilities of the different surfaces to trap electrons. Thus, it might be expected that the possibility of adsorbing a second layer of Hg⁰ also followed the abovementioned order, as the greater electron deficiency favored electron transfer from Hg⁰ to the surface.

As shown in Fig. 4(a) and (b), the second layer of Hg adsorbed on the intact ZnS (111) surface exhibited an adsorption energy of only 28.2 kJ mol⁻¹, which indicated physical adsorption and a tendency to desorb at the reaction temperature. On the S-defect and Zn-defect surfaces, the second-layer of Hg nevertheless exhibited adsorption energies of 45.4 and 98.5 kJ mol⁻¹, much greater than those calculated for adsorption on the intact ZnS (111) surface; this indicated more resistance to the higher reaction temperatures. It should be noted that these DFT calculation results were fully in line with the experimental Hg-TPD patterns for different samples, as shown in Supplementary Fig. 13. The highlighted areas in the Hg-TPD patterns for 0.368-ZnS indicated that new active sites emerged after etching by acids. The area indexing to a higher desorption temperature was ascribed to more

stable surface immobilization with higher adsorption energy, while the Hg⁰ desorbed at ~150 °C might be attributed to multilayer adsorption of mercury. Therefore, according to the DFT calculations and Hg-TPD results, the enhanced Hg⁰ capacity of defect-rich ZnS without the influence of mass transfer resulted from the increased number of active sites favoring multilayer adsorption. This speculation was also supported by the STEM-HAADF images of Hg-laden pristine and acid-etched ZnS (as shown in Fig. 4(c) and (d)). In the Hg-laden pristine ZnS, no obvious amorphous layer was identified around the nanoparticles, while the acid-etched ZnS was encapsulated by an obvious amorphous shell attributed to multilayer adsorption of Hg. It should be noted that in both fresh pristine and acid-etched ZnS, no amorphous layer was found encapsulating the ZnS nanoparticles (as shown in Supplementary Fig. 20), which also indicated that the amorphous layer in Hg-laden 0.368-ZnS was formed during the Hg⁰ adsorption process.

After determining mechanistically how and why acid etching increased the abundance of active sites in ZnS, a fixed-bed reaction system was used to include the influence of Hg⁰ mass transfer and simulate real-world conditions (as shown in Supplementary Fig. 3 and Fig. 5(a-h)). As shown in Fig. 5(a), 0.368-ZnS exhibited optimal Hg⁰ adsorption at 100–150 °C. The inadequate Hg⁰ capture efficiencies exhibited at temperatures below 100 °C were probably caused by insufficiently activated adsorption sites, while those seen at high temperatures above 150 °C might be attributed to desorption/

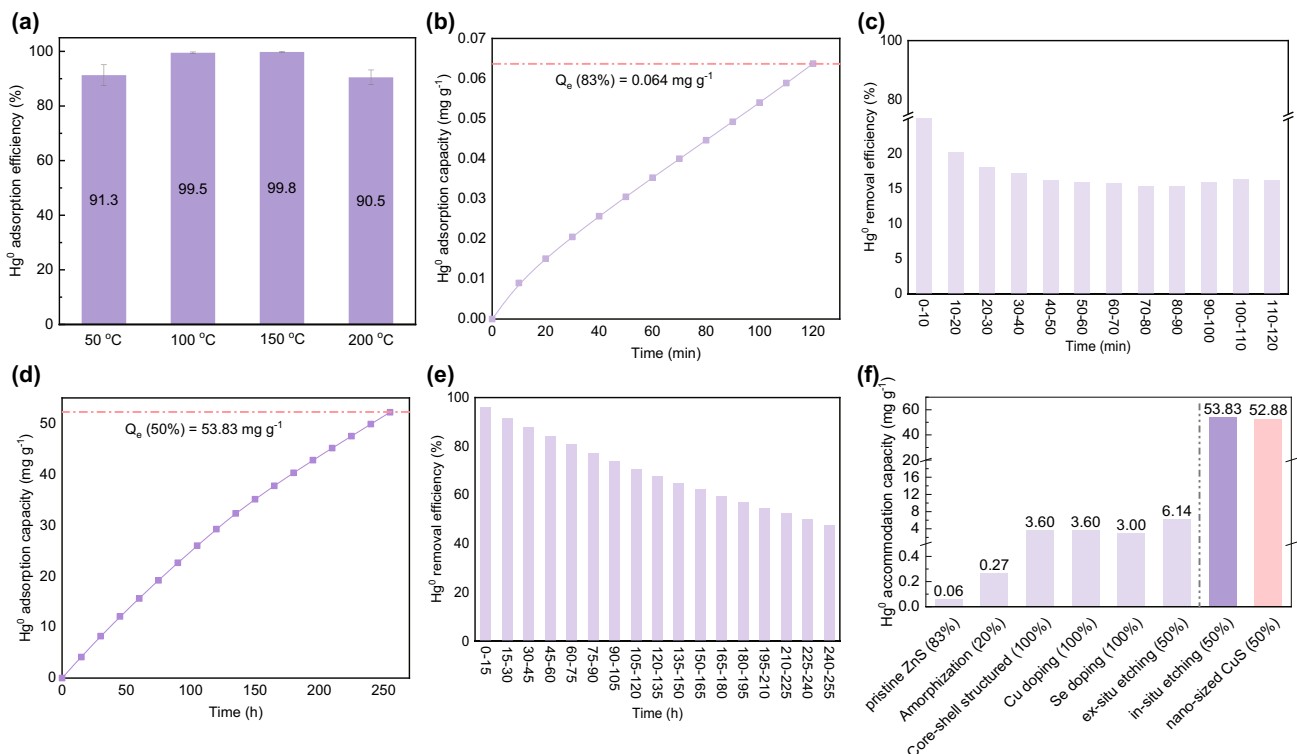

**Fig. 5 | Hg⁰ adsorption performances of different ZnS samples. a** Influence of reaction temperature on the Hg⁰ removal performance of 0.368-ZnS; the Hg⁰ accommodation capacities and real-time Hg⁰ removal efficiencies of (**b**) & (**c**) pristine ZnS and (**d**) & (**e**) 0.368-ZnS; and (**f**) a Hg⁰ accommodation capacity comparison among different ZnS-based sorbents and nanosized CuS. The error bars in panel (**a**) correspond to the standard deviation of three independent measurements, and the data in panel (**a**) are presented as mean values of three independent measurements.

decomposition of the adsorbates[36]. A temperature of 100 °C was thus used to test the Hg⁰ accommodation capacities of the ZnS-based sorbents and ensure optimization of both the pristine and acid-etched ZnS[7,24]. As shown in Fig. 5(b) & (c), at the beginning of the breakthrough experiment, 20 mg of pristine ZnS only exhibited an ~30% Hg⁰ removal efficiency, and it had further decreased to 17% when the experiment ended. The Hg⁰ accommodation capacity was 0.064 mg g⁻¹, generally in line with the value reported in previous research[24]. Through in situ etching, the Hg⁰ accommodation capacity of ZnS increased to 53.83 mg g⁻¹, and at this capacity value, 20 mg of 0.368-ZnS still maintained ~50% of the Hg⁰ adsorption efficiency after the experiment lasted for 250 h (as shown in Fig. 5(d) & (e)). In addition, similar to what occurred in the nested-tube reaction system, ex0.368-ZnS exhibited a moderate Hg⁰ capture capacity of 6.14 mg g⁻¹, which was between those of pristine ZnS and 0.368-ZnS (as shown in Supplementary Fig. 21 for a breakthrough threshold of 50%), further suggesting that the ex situ-etched ZnS combined the properties of both pristine ZnS and in situ-etched ZnS. The Hg⁰ accommodation capacity of 0.368-ZnS, which was higher by approximately three orders of magnitude compared to that of pristine ZnS, indicated that, in addition to an abundance of active sites, its porous structure also played a fundamental role in aiding Hg⁰ adsorption with enhanced transport and diffusion.

The Hg⁰ accommodation capacities of different ZnS-based sorbents, including pristine, amorphized, core-shell structured, heteroatom doped, ex situ etched, and in situ etched sorbents, were compared with that of nanosized CuS, a benchmark metal sulfide exhibiting a capacity that is industrially applicable. As shown in Fig. 5(f), pristine ZnS, amorphized ZnS, core-shell structured ZnS, Cu doped, Se doped, and ex situ etched ZnS exhibited Hg⁰ accommodation capacities of 0.06 (83%), 0.27 (20%), 3.60 (100%), 3.60 (100%), 3.00 (100%), and 6.14 (50%) mg g⁻¹, respectively[6], for which

the numbers in the parentheses were the breakthrough thresholds at the time when the adsorption capacities of the designated sorbents were calculated. Although the breakthrough thresholds of amorphized ZnS and ex situ etched ZnS were not close to 100%, their capacities far below the capacity of 50%-penetrated CuS suggested limited prospects for practical use. In contrast, when the breakthrough threshold of 0.368-ZnS was identical to that of nanosized CuS, they shared similar Hg⁰ accommodation capacities (53.83 and 52.88 mg g⁻¹), suggesting that the equilibrium Hg⁰ adsorption capacity of 0.368-ZnS might approach the critical value of 0.1 g g⁻¹, which would successfully and significantly extend the applicability of ZnS in Hg⁰ remediation.

Typical components present in different industrial flue gases have been reported to inhibit Hg⁰ adsorption by metal sulfides, and these were evaluated for their effects on Hg⁰ capture by pristine ZnS and 0.368-ZnS (as shown in Fig. 6(a) & (b)). Both SO₂ and H₂O slightly inhibited Hg⁰ adsorption on pristine ZnS, probably because of competition for active sites among Hg⁰, SO₂, and H₂O[12,16,22] and/or coverage of active sites by the highly concentrated H₂O[15,37]. O₂ gas is a highly enriched component in industrial flue gases, but it exhibits a negligible influence on Hg⁰ adsorption by metal sulfides[10,15,38–40], so it was not studied in this work. After treatment via in situ etching, the competitive adsorption effect was partially offset, suggesting that 0.368-ZnS might have a higher selectivity for adsorption of Hg⁰ than for the flue gas components (which may probably be ascribed to the increased stability of mercury on the in situ etched sample). Finally, the influence of acid species on Hg⁰ removal by ZnS was also tested (as shown in Fig. 6(c)), and the results showed comparable effectiveness for H₂SO₄, nitric acid (HNO₃), and hydrochloric acid (HCl) in enhancing the Hg⁰ adsorption capacity of ZnS. However, acetic acid (HAC) compromised the Hg⁰ adsorption capacity of ZnS, which was probably attributable to its weak acidity, but the specific reason for this phenomenon remains

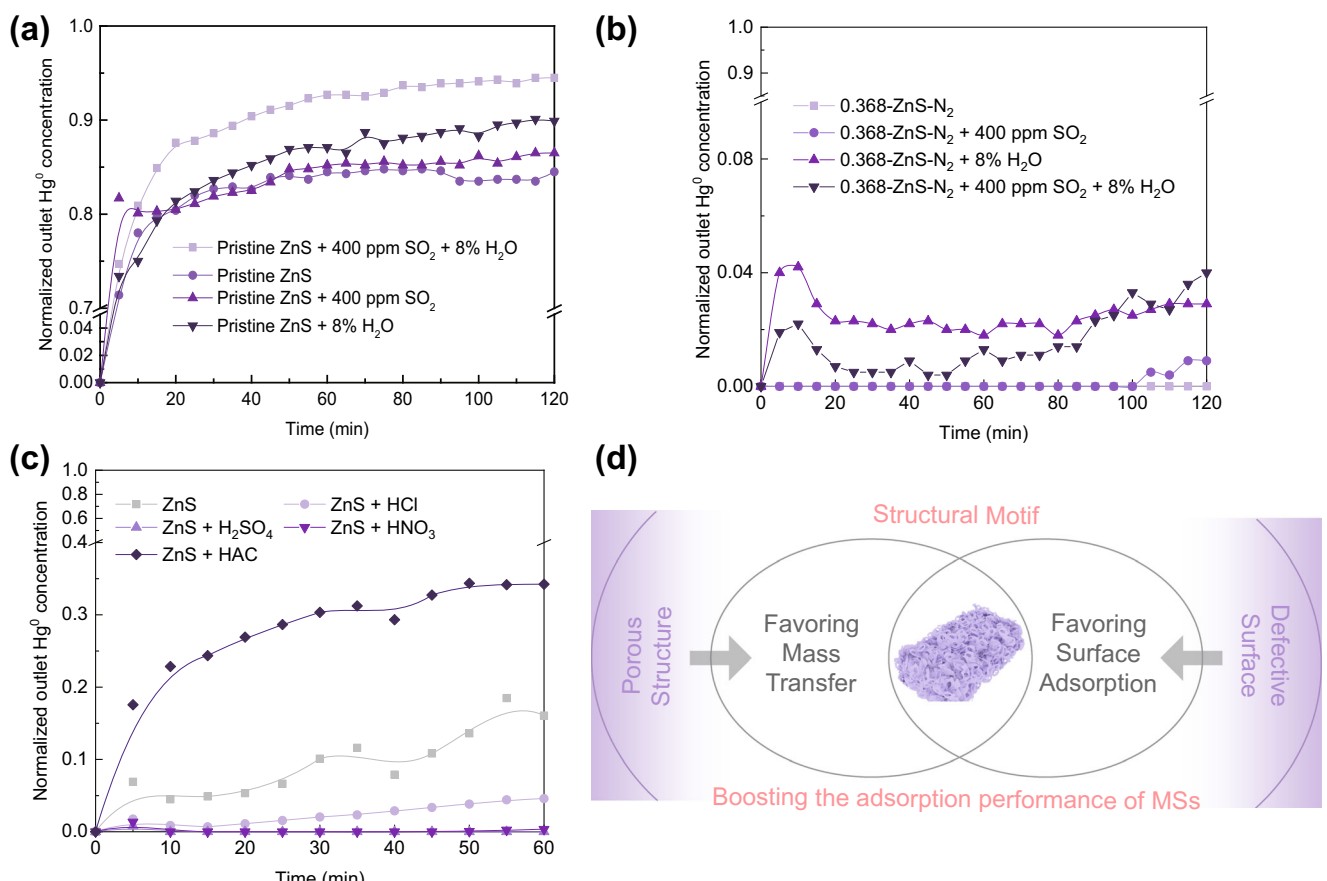

**Fig. 6 | Influences of different factors on Hg⁰ adsorption performances.** The influence of flue gas components on the Hg⁰ capture behaviors of pristine (**a**) ZnS and (**b**) 0.368-ZnS; (**c**) the effects of acid species on the Hg⁰ capture performance of acid-etched ZnS; and (**d**) a diagrammatical illustration of the structural motif of acid-etched ZnS.

to be further explored and elucidated. A structural motif comprising porous architectures and surface defects was hence proposed to result in the boosted Hg⁰ adsorption behaviors of the TMSs under practical scenarios (as illustrated by Fig. 6(d)).

After the related mechanisms were elucidated, extensive application of the in situ etching method was further established for different TMSs. As shown in Fig. 7(a-d), four TMSs, including CoS, NiS, CdS, and MoS₂, were taken as typical examples to probe the generality of this efficient method. Compared with the pristine samples, acid-etched CoS, NiS, CdS, and MoS₂ all exhibited enriched porosities, indicating that in situ acid etching affected other the TMSs as it did ZnS, i.e., by constructing a structural motif that benefitted heterocomponent adsorption. Among these sorbents, it should be noted that MoS₂ was synthesized via a hydrothermal pathway, suggesting that functionalization with the in situ etching method was also possible under relatively high temperatures and pressures. The Hg⁰ adsorption behaviors of pristine and acid-etched CoS, NiS, CdS, and MoS₂ are shown in Fig. 7(e) and Supplementary Fig. 22. The pristine CoS, NiS, CdS, and MoS₂ exhibited 50% Hg⁰ adsorption capacities of 0.58, 0.54, 0.33, and 0.18 mg g⁻¹, respectively; these were at least two orders of magnitude lower than that of CuS and hence were not applicable under practical scenarios. However, after being subjected to in situ etching, the 50% Hg⁰ adsorption capacities of CoS, NiS, CdS, and MoS₂ increased by two magnitudes to 74.1, 56.1, 34.1, and 34.5 mg g⁻¹, and all approached, or even exceeded, that of CuS under the same reaction conditions. This observation demonstrated clearly that the in situ etching method was broadly general in boosting the TMSs capacities for Hg⁰ and heterocomponent adsorption.

Considering the facile procedure and high performance of the in situ acid etching method, it may significantly extend the applicability of metal sulfides for heterocomponent adsorption, as mentioned above. Hg⁰ is a typical example, for which removal is currently a global concern but lacks flexible and versatile countermeasures. For the effective capture of Hg⁰ from industrial flue gases, traditional activated carbons have been gradually replaced by metal sulfides because they contain abundant sulfur ligands, which are Lewis acid, that stably adsorb Lewis bases such as Hg⁰[11]. However, due to the unfavorable coordination environment and pore structure, most of the sulfur ligands in metal sulfides are not available for Hg⁰ accommodation. After treatment by the in situ acid etching process, the sulfur ligands in most metal sulfides can be effectively activated, which offers more options for abating anthropogenic Hg⁰ pollution and recycling the mercury resources. For instance, in removal of Hg⁰ from various industrial processes, such as coal combustion, waste incineration, and cement production, the operating temperatures are generally higher than 120 °C[7,41–43], and the Hg⁰ removal performance of CuS is largely compromised. With application of the in situ acid etching method, sorbents such as ZnS and CdS that exhibit Hg⁰ adsorption capacities comparable to that of CuS within the temperature range 120 to 200 °C will be feasible alternatives to activated carbons in the future. In addition, geological access to different metal precursors and the preferences of different industries can be taken into consideration when choosing the optimal technology for Hg⁰ removal and recovery. Considering the mercury loading capacities of in situ etched TMSs approach, or even exceed the mercury loading capacities of the natural sulfide ores that has widely and long been (generally ranging between

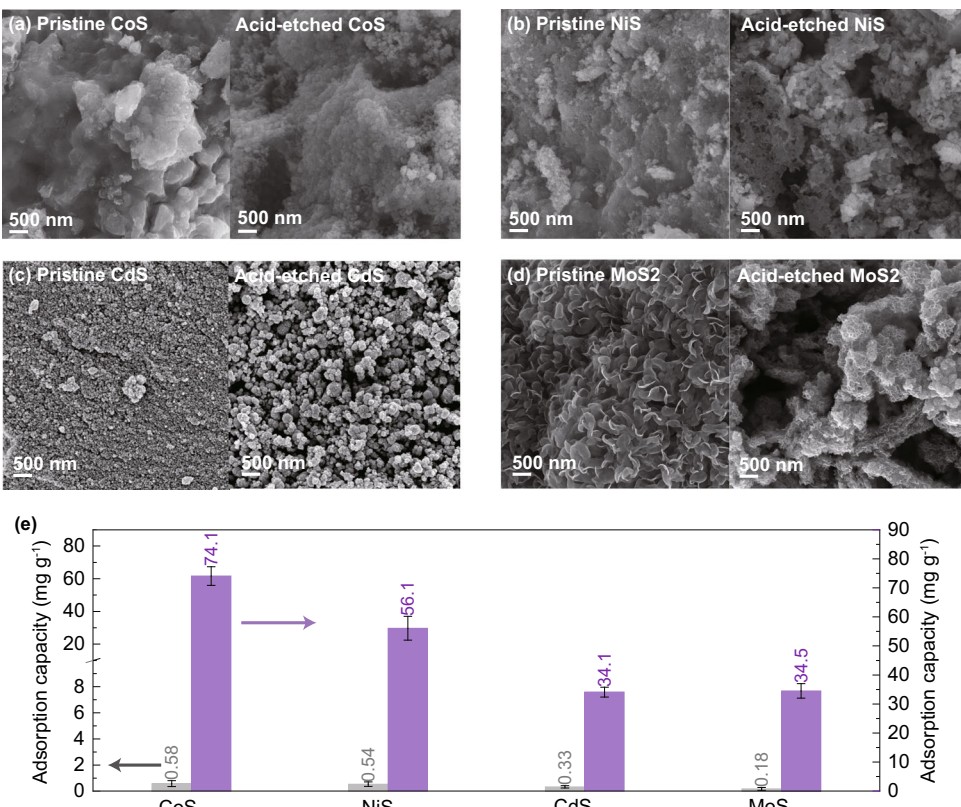

**Fig. 7 | Hg⁰ adsorption performances of different TMSs after acid etching.** SEM images of pristine and acid-etched (**a**) CoS, (**b**) NiS, (**c**) CdS, and (**d**) MoS₂; and (**e**) Hg⁰ removal efficiencies of pristine and acid-etched CoS, NiS, CdS, and MoS₂. The error bars in panel (**e**) correspond to the standard deviation of three independent measurements, and the data are presented as mean values.

0.3–10%) used for mercury smelting and production[44], extracting metal mercury from Hg-laden acid etched TMSs will be cost-effective and industrially-practical, e.g., the metal mercury recycled from Hg-laden 0.368-ZnS was shown in Supplementary Fig. 23. Finally, the adsorption behaviors of heterocomponents such as $NH_3$ and CO, which are Lewis bases like Hg⁰, are also expected to be significantly improved by in situ acid etching of metal sulfides. This prospect may further extend the opportunities for application of this synthetic method to catalytic energy conversion and storage.

In summary, this work was designed to develop an in situ acid-etching method to extend the range of applicable TMSs available for Hg⁰ capture from polluted sources. ZnS was taken as a typical example with which to explore the mechanistic fundamentals responsible for the capacity improvements, showing that the increased abundance of active sites and the optimized architecture derived from the porous structure constituted a favorable motif for Hg⁰ sequestration. Among the characteristics induced by the stepwise dissolution of TMSs during the in situ etching process, the undercoordinated surface sites favored the multilayer adsorption of Hg⁰, and the porous architecture of the TMS framework promoted the diffusion of Hg⁰. The Hg⁰ accommodation capacity of ZnS (with a 50% breakthrough threshold) was significantly improved from 0.064 mg g⁻¹ to 53.83 mg g⁻¹ after etching and approached the critical value exhibited by CuS (52.88 mg g⁻¹). Compared to other modification methods reported in previous studies, including amorphization, core-shell structuring, and heteroatom doping, in situ acid etching was at least ten times more effective. Using this effective method, acid-etched CoS, NiS, CdS, and MoS₂ were also synthesized, and their Hg⁰ accommodation capacities were increased from 0.58, 0.54, 0.33, and 0.18 mg g⁻¹ to 74.1, 56.1, 34.1, and 34.5 mg g⁻¹, respectively, which supported the generality of the in situ acid-etching method in boosting the Hg⁰ adsorption capacities of TMSs. From these

perspectives, it is reasonable to conclude that in situ acid etching will significantly extend the range of TMSs applicable for Hg⁰ pollution abatement and even other environmental remediation and energy conversion processes involving comparable (pre)adsorption of heterocomponents.

## Methods
### Materials
The raw materials used in this work including zinc sulfate (heptahydrate, $ZnSO_4 \cdot 7H_2O$, 99.5%), sodium sulfide (nonahydrate, $Na_2S \cdot 9H_2O$, 99.99%), cobalt chloride (hexahydrate, $CoCl_2 \cdot 6H_2O$, 99.9%), nickel sulfide (hexahydrate, $NiSO_4 \cdot 6H_2O$, 99.9%), cadmium sulfide (8/3-hydrate, $3CdSO_4 \cdot 8H_2O$, 99.99%), ammonium molybdate (tetrahydrate, $H_{24}Mo_7N_6O_{24} \cdot 4H_2O$, > 99.0%), thiourea ($CH_4N_2S$, 99.0%), sulfuric acid ($H_2SO_4$, 98%), nitric acid ($HNO_3$, 70%), hydrochloric acid (HCl, 36.5%), acetic acid ($CH_3COOH$, Analytical Grade) were purchased from Aladdin.

### Sorbent synthesis
In situ acid-etched ZnS was synthesized by a precipitation method. Typically, $ZnSO_4 \cdot 7H_2O$ (0.01 mol) was dissolved in a 0.368 mol L⁻¹ acid solution ($H_2SO_4$ was used in most cases that are not specially noted), and in a separated beaker, a stoichiometric $Na_2S$ solution was prepared. After two homogeneous solutions were obtained, they are mixed together, hence a light-yellow precipitate formed immediately. The precipitate was repeatedly washed by deionized water and ethanol for several times, dried in vacuum at 60 °C for 12 h, and grounded through 200 meshes to be 0.368-ZnS. By changing the concentration of acid, 0.0184-ZnS, 0.092-ZnS, and 0.184-ZnS were synthesized with the prefix number denoting the concentration of acid. Pristine ZnS was prepared through a same procedure without the addition of any acid.

Ex-situ acid-etched ZnS was obtained via impregnating pristine ZnS in $H_2SO_4$ (0.368 mol $L^{-1}$). Besides, a similar process was adopted to prepare in-situ acid-etched CoS, NiS, and CdS, while $MoS_2$ was synthesized based on a hydrothermal pathway with the addition of $H_{24}Mo_7N_6O_{24}\cdot4H_2O$ and $CH_4N_2S$ in the presence of $H_2SO_4$. For the hydrothermal preparation of pristine or acid-etched $MoS_2$, $H_{24}Mo_7N_6O_{24}\cdot4H_2O$ (2.47 g) was dissolved in deionized water or $H_2SO_4$ solution (0.368 mol $L^{-1}$), and then $CH_4N_2S$ (2.14 g) was added. After stirring for 10 minutes, the mixed solution was transferred into a 100 ml Teflon-lined stainless-steel autoclave and maintained at 220 °C for 18 h. Subsequently, the reaction system was cooled to room temperature, and the precipitate was repeatedly washed by deionized water and ethanol for several times and dried in vacuum at 60 °C for 12 h.

## Sorbent characterization

The crystallinity of the sorbents was measured by X-ray diffraction (XRD, D8 Bruker AXS, Germany) with two thetas from 10° to 80° in $Cu_\alpha$ (λ = 0.154 nm) radiation. The morphologies of the sorbents were recorded by a scanning electronic microscope (SEM, FEI F50, USA), during which the energy-dispersion X-ray (EDX) scanning and mapping were conducted to determine the composition content and distribution. A transmission electron microscope (TEM, JEOL 2100 F, Japan) was used to determine the morphologies of the as-prepared sorbents. Besides, a scanning-transmission electron microscope (STEM, Titan G2 60–300), equipped by a high-angle annular dark-field detector (HAADF), was used to further obtain the atom-level understanding of sorbents. The TEM and STEM-HAADF results were analyzed with the Digital Micrograph DM3 software. The X-ray absorption fine structure spectra (Zn K-edge) were collected in Beijing Synchrotron Radiation Facility (BSRF). The storage rings of BSRF were operated at 2.5 GeV with an average current of 250 mA. Using Si (111) double-crystal monochromator, the data collection was carried out in transmission/fluorescence mode using ionization chamber. All spectra were collected in ambient conditions. The acquired EXAFS data was processed according to the standard procedures using the ATHENA module implemented in the IFEFFIT software packages. The $k^3$-weighted EXAFS spectra were obtained by subtracting the post-edge background from the overall absorption and then normalizing with respect to the edge-jump step. Subsequently, $k^3$-weighted χ(k) data of Zn K-edge were Fourier transformed to real (R) space using a Hanning windows (dk = 1.0 Å$^{-1}$) to separate the EXAFS contributions from different coordination shells. To obtain the quantitative structural parameters around central atoms, least-squares curve parameter fitting was performed using the ARTEMIS module of IFEFFIT software packages (0.9.26). The inductively coupled plasma mass spectrometry (ICP-MS, X Series 2) was adopted to determine the compositional ratios of Zn and S in different ZnS samples. The ZnS samples were dissolved under alkaline conditions to avoid the evaporation of $H_2S$. The fresh and spent sorbents were characterized by their X-ray photoelectron spectroscopy (XPS, Thermo ESCALAB 250Xi) spectra with a C 1 s binding energy value of 284.8 eV as the reference. The spent sorbent was prepared by pre-adsorbing $Hg^0$ over the metal selenide surface. The Brunauer-Emmett-Teller (BET) surface area of the sorbents was determined by the $N_2$ adsorption and desorption method with a BET analyzer (ASAP 2020, Micromeritics, USA). Before BET testing, the prepared sorbents were purged in pure $N_2$ for 4 h to obtain a clean surface.

## Activity test

The equilibrium $Hg^0$ adsorption capacities of sorbents were evaluated by a fixed-bed reaction system (as shown in Supplementary Fig. 3). Compressed gas cylinders containing nitrogen ($N_2$) was used to introduce carrier gas and $Hg^0$ into the reaction system. The total gas flow rate was precisely controlled with a mass flow controller to be 1 L $min^{-1}$. Stable vapor-phase $Hg^0$ (100 μg $m^{-3}$) was provided with a $Hg^0$

permeation device (Dynacal, VICI Metronics) through heating the permeation tube to a constant temperature. A reactor made of borosilicate glass with an inner diameter of 10 mm was placed in a tubular furnace equipped with a controlled temperature variation of less than 2.0 °C. The $Hg^0$ concentration was continuously recorded with a $Hg^0$ analyzer (VM3000, Mercury Instrument, Inc.). The exhausted gas passed through a mercury trap containing CuS prior to discharge. Before each test, the gas flow bypassed the reactor loaded with sorbents until the $Hg^0$ concentration fluctuation was lower than 1 μg $m^{-3}$ for 30 min. The $Hg^0$ concentration in the bypass air was denoted as the inlet $Hg^0$ concentration ($C_{in}$). After that, the gas flow passed through the sorbents and the detected $Hg^0$ signal was designated as the outlet $Hg^0$ concentration ($C_{out}$). The normalized $Hg^0$ concentration, the real-time $Hg^0$ adsorption capacities ($Q_t$, mg $g^{-1}$), and the $Hg^0$ removal efficiency (η, %) of the sorbents were calculated using Eqs. (4−6):

$$\text{Normalized } Hg^0 \text{ Concentration} = \frac{C_{out}}{C_{in}} \tag{4}$$

$$Q_t = \frac{1}{m} \int_{t_1}^{t_2} (C_{in} - C_{out}) \times f \times dt \tag{5}$$

$$\eta = \frac{\int_{t_1}^{t_2} (C_{in} - C_{out}) \times f \times dt}{\int_{t_1}^{t_2} C_{in} \times f \times dt} \tag{6}$$

where $f$ ($m^3$ $min^{-1}$) is the gas flow rate, $m$ (g) is the mass of the sorbent, and $t$ (min) is the duration time of reaction.

Besides, to rule out the effect of favorable pore structure and mass transfer, a nested-tube reactor was also used in this work to exclusively characterize the abundance of active sites on the surface of different sorbents (as shown in Supplementary Fig. 12). It is well-known that, in a nested-tube reactor, the concentration of $Hg^0$ approaches infinity and the adsorption rate far exceeds the desorption rate, making the desorption rate ignorable and the mass transfer irrelevant. In a typical process, The $Hg^0$ source was placed in the bottom of the inner reactor without a lid, while the sorbent was settled on the top of the inner reactor and separated by a filter paper that was capable of permeating $Hg^0$ but inert for $Hg^0$ adsorption. Then, the $Hg^0$ source in the sealed tube reactor was heated by an oil bath at 140 °C and held for 1 d, and the $Hg^0$ concentration reached nearly infinite in a sealed tube reactor. The amount of adsorbed mercury was determined by a mercury temperature programmed desorption/decomposition (Hg-TPD) test. Before the test began, the Hg-laden samples were purged by pure $N_2$ until the outlet $Hg^0$ concentration equaled to zero. The Hg-TPD test was subsequently conducted with a heating rate of 5 °C $min^{-1}$, and the on-line mercury analyzer was adopted to record real-time $Hg^0$ concentration during this process.

## DFT calculation

To investigate the $Hg^0$ adsorption behaviors on intact, S-defect, and Zn-defect surfaces, DFT calculations were conducted. A ZnS model with the space group of F-43m (a = b = c = 5.45 Å, α = β = γ = 90°) was used according to the XRD characteristic results (as shown in Supplementary Fig. 2). The exposure ratios of different ZnS crystal facets were determined by the Morphology Tools in Materials Studio 2020 with the assistance of classic BFDH model. A 3 × 3 slab model comprised of five layers along the <111> direction was constructed based on the surface energies of different crystal facets. A 20.0 Å vacuum region between the slabs was constructed to avoid the spurious interactions. The S-defect or Zn-defect surface was simulated by removing one of surface S or Zn atom off. All calculations were conducted with the quantum mechanics based CASTEP program package in Materials Studio 2020[45]. The ultrasoft pseudopotentials

(USP) proposed by Vanderbilt was adopted to describe the electron-ion interactions[46]. The exchange correlation potential was determined by Perdew-Burke-Ernzerhoff approximation for solid (PBEsol) in the place of the generalized gradient approximation (GGA) scheme[47]. The Monkhorst-Pack scheme k points grid of $2 \times 2 \times 1$ was used to simplify the Brillouin zone, and a cutting off energy of 400.0 eV was used. The criteria for the tolerances of energy, force, displacement, and SCF convergence criteria are set as $10^{-5}$ Ha, $2 \times 10^{-2}$ Ha Å$^{-1}$, $10^{-3}$ Å, and $10^{-5}$, respectively. The cell parameters of ZnS after geometry optimization were $a = b = c = 5.39$ Å, deviating from the standard pattern by 1.1%, proving the feasibility to adopt this method for DFT calculations. Besides, the Hg$^0$ atom was also separately optimized through a same method in a $10 \times 10 \times 10$ Å vacuum box. The adsorption energy ($E_{ad}$, kJ mol$^{-1}$) of Hg$^0$ can hence be obtained by Eq. (7):

$$E_{ad} = E_{sorbent-adsorbate} - (E_{sorbent} + E_{adsorbate}) \qquad (7)$$

where $E_{adsorbate-sorbent}$ (kJ mol$^{-1}$) represented the total energy of the adsorbate-sorbent system, and $E_{adsorbate}$ (kJ mol$^{-1}$) and $E_{sorbent}$ (kJ mol$^{-1}$) were the energy of isolated adsorbate and clean sorbent. From Eq. (4), it can be concluded that more negative the $E_{ad}$ was, stronger the adsorption exhibited, while positive E$_{ad}$ indicated that the specific adsorbates could not adsorb on the ZnS(111) surface. All figures in this work were drawn in Originlab 2018 software.

### Statistics and Reproducibility
For all experiments, particularly for the results as shown in Figs. 1(a–c), 2(a–c), 4(c–d); Supplementary Figs. 7(a–c), 9, 11, and 20(a–b), the data were sampled according to the minimal number of independent replicates that significantly identified an effect (repeating at least three times). Repeated measurements of the evolving quantities showed deviations of less than 10% confirming reproducibility of the reported experiments.

### Reporting summary
Further information on research design is available in the Nature Portfolio Reporting Summary linked to this article.

## Data availability
The data that support the finding of this study are contained in the Article and its Supplementary Information file. Source Data are provided with this paper. Additional raw data are available from the corresponding author upon request. Source data are provided with this paper.

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

## Acknowledgements

This project was supported by the National Natural Science Foundation of China (No. 22106183 - Z. Y., 52276144 - H. L., 52176145 – W. Q.), National Key Research and Development Program of China (No. 2022YFC3701505-04 - H. L.), Science and Technology Innovation Program of Hunan Province (No. 2021RC4005 - H. L.), Natural Science Foundation of Hunan Province (No. 2022JJ40575 - Z. Y., 2021JJ30851 - J. Y.), and Open Sharing Fund for the Large-scale Instruments and Equipment of Central South University (Z. Y.).

## Author contributions

H.L. and Z.Y. conceived and supervised the research and wrote the manuscript. J.Z. and W.Z. designed and conducted the experiments. H.Z. and H.C. conducted the DFT calculations. J.Y., W.Q., L.L., and Y F. participated in characterizations and data analyses. All authors discussed the results and commented on the manuscript.

## Competing interests

The authors declare no competing interests.
