## [Peer Review File · Nature Communications]

Reviewer comments, first round review –

Reviewer #1 (Remarks to the Author):

Hg⁰ capture by metal sulfides rises as an interesting research topic in recent years. It is highly appreciated to develop such a novel pathway to boost the Hg⁰ accommodation capacities of metal sulfides. Based on the novelty and applicability of this pathway, I draw an endorsing conclusion for its publication on Nat. Comm with minor revision. There are several concerns remain to be addressed.

- (i). The normalized Hg⁰ concentration mentioned in line 116-117 should be explained.
- (ii). In line 177-178, it is said that 'the structural voids created more edge sites...'. For the statement in this sentence, literature supports should be provided to prove its correctness and validation.
- (iii). The peak location change in Figure 2(d) cannot be clearly read in its current version. It is suggested to exclusively show only the Zn 2p 3/2 peak in Figure 2(d) because the Zn 2p 3/2 and 2p 1/2 peaks moved along the x axis with the same value. Thus, no need to include two peaks in this figure.
- (iv). Figure 2(h) is insufficient to show both the formation of structural and atomic defects in acid-etched ZnS. The authors may consider to redraw this schematic figure to serve a better readability.
- (v). Figure S13 only showed the lattice fringe along different orientations of the ZnS crystal. This did not agree with the discussions in line 223-225. If the exposure probability of different facets was obtained based on their Miller indices? The compatibility between the figure content and the discussions should be further checked.
- (vi). The STEM-HAADF that visualized the edge areas of both pristine and acid-etched ZnS should be added in the supporting information. The main goal is to prove that the amorphous layer outside the nanoparticle as shown in Figure 3(f) & (g) formed after the Hg⁰ adsorbed on ZnS.
- (vii). What the numbers in the bracelets in line 316 denotes should be specified.
- (viii). In line 325-327, the inhibitive effects of SO₂ and H₂O were attributed to the competitive adsorption between SO₂/H₂O and Hg⁰. More literature evidences should be provided to support this speculation. Besides, if there is any other reason may explain the inhibitive effects of SO₂ and H₂O?
- (ix). The authors only considered the influence of SO₂ and H₂O on the Hg⁰ capture performances of ZnS. Why the impact of O₂, a flue gas component widely existing in flue gases, was not evaluated?
- (x). The specific experimental conditions for the preparation of MoS₂ should be elaborated. For example, at what temperature this sample was hydrothermally treated, for how long time, and so on.
- (xi). The Hg⁰ breakthrough curves for figure 5(e) should be provided in supporting information.
- (xii). Insufficient evidence was provided to prove that CuS possesses the highest Hg⁰ adsorption capacity among all metal sulfides. A comparison table showing the Hg⁰ adsorption capacities of different metal sulfides may be presented in the supporting information.

Reviewer #2 (Remarks to the Author):

This work proposed a very efficient and meaningful method to overcome one of the major challenges impeding the application of most metal sulfides for Hg⁰ remediation. Amazing outcomes showed that the Hg⁰ adsorption capacities of all metal sulfides pretreated by in-situ etching method

reached the level of tens of milligram per gram, approaching or surpassing the critical value defined by CuS. The meaningfulness and contribution of this valuable work indicates its suitability to be published on Nature Communications after some minor revisions are made.

1. The in-situ dissolution of samples as shown by Figure 1(g) is valuable outcomes supporting the boosted Hg₀ adsorption performance of 0.368-ZnS. Thus, it is suggested to further expand discussions about the influences of the in-situ and ex-situ dissolution process on the textural property of ZnS and its Hg₀ removal performance.
2. The weight loss of ZnS in etching process with 0.368 M is about 50%, now I wonder how about the ZnS yields in other conditions, e.g. 0.092 and 0.184 M. As the author said the Hg₀ adsorption performance and yield rate of ZnS should be purposefully traded off under practical scenarios. So you need to provide the critical value on this issue.
3. The resolutions of XPS deconvolution patterns needs to be improved. The curves colors for Figure 2 (g) needs to be adjusted to make readers easily distinguish their difference. XPS results of sorbents after adsorbing Hg₀ may be added to further support the interaction between ZnS and Hg₀.
4. Could ZnSO₄, a possible product after adding H₂SO₄, be washed out and its role be fully ruled out? Although the XRD results showed no peaks belonging to ZnSO₄, it is possible that the peaks of ZnSO₄ overlapped with the broad peaks of ZnS. In the S 2p XPS image, signals beyond 166 eV were not presented. It is hard to judge if there was ZnSO₄ left in the sample (XPS signal locating at around 168 eV). It is suggested to provide S 2p signal at 168 eV to rule out the influence of ZnSO₄.
5. Can any application implication be derived from the amazing theoretical outcomes of this manuscript? A paragraph discussing the application implication of these outcomes may be added before the conclusion is derived.
6. The calculation on the exposure ratios of different crystal surfaces is interesting and meaningful, what is the calculation formula, I recommend the authors providing it or the relevant reference.
7. The authors only showed the common analysis results to support the interaction between Hg and ZnS in the DFT calculations. It is good but not enough. Partial density orbital states (PDOS) may be a better method to elucidate this interaction from electronic level. Thus, PDOS results are suggested to be added.
8. The resolutions of all panels in Figure 4 needs to be improved.
9. What was the percent of under-coordinated sulfur in CuS? Did CuS form hierarchical porous structure to promote the Hg₀ transportation? In other word, whether the structural properties of CuS were inherited by the acid-etched ZnS to make the assumption in the introduction section reasonable?
10. Figure S17 is a fundamental one that explains the roles of structural motif in the Hg₀ adsorption by ZnS. Such significance makes this figure more suitable in the main content instead of supporting information.
11. The authors are recommended to check the consistency of the tense of verbs used in this manuscript.

Reviewer #3 (Remarks to the Author):

The mercury capture by transition metal sulfides (TMSs) is supposed to be potentially applicable in industrial flue gas cleaning and mercury-laden waste decontamination, acting as potential alternatives to activated-carbon-based techniques considering the cost-effectiveness and eco-friendliness. This paper presents the synthesis of the one-step in-situ acid-etching method. The authors mixed mild acids with TMS throughout the sample preparation process. ZnS was taken as a model material to explore the mechanistic fundamentals responsible for the capacity improvement,

evidencing that the increased abundance of surface-active sites and the optimized architecture composed of the porous structure formed a favorable motif that served the overall benefits of Hg⁰ sequestration. Considering the impact and quality, this manuscript seems to be timely and contains important information for academic and industrial researchers. Hence, I support its acceptance by Nature Communications. However, some issues need to be addressed prior to publication. The authors should discuss more on the critical issues in the field in the introduction section. Furthermore, detailed structural characterization of samples, such as FFT analysis from the HRTEM images, and ICP-MS data, are highly required to verify the defective crystal structures of as-prepared samples. In addition, supporting evidence from ensemble analysis with atomic-scale element mappings, EXAFS, and EPR study would provide a undeniable evidence for the presence of the defects. Besides, the author should carefully check the whole manuscript. There are several grammar mistakes.

Reviewer #1

Hg⁰ capture by metal sulfides rises as an interesting research topic in recent years. It is highly appreciated to develop such a novel pathway to boost the Hg⁰ accommodation capacities of metal sulfides. Based on the novelty and applicability of this pathway, I draw an endorsing conclusion for its publication on Nat. Comm with minor revision. There are several concerns remain to be addressed.

(i). The normalized Hg⁰ concentration mentioned in line 116-117 should be explained.

We thank the reviewer for this great reminder. The normalized Hg⁰ concentration as mentioned has been defined in the revised supporting information (please refer to the 1st paragraph in page S4). The same equation to calculate the normalized Hg⁰ concentration is shown below for your reference, where C_{in} (μg m⁻³) and C_{out} (μg m⁻³) represent the inlet and outlet Hg⁰ concentrations.

$$\text{Normalized Hg}^0 \text{ Concentration} = \frac{C_{out}}{C_{in}}$$

(ii). In line 177-178, it is said that ‘the structural voids created more edge sites...’. For the statement in this sentence, literature supports should be provided to prove its correctness and validation.

We thank the reviewer for this constructive comment. Supportive references have been added in the revised manuscript to prove the correctness and validation of the abovementioned statement. The list of added references is - (1) Kibsgaard J, et al. Engineering the surface structure of MoS₂ to preferentially expose active edge sites for electrocatalysis. Nat. Mater. 11, 963-969 (2012); (2) Wang Z, et al. Controllable etching of MoS₂ basal planes for enhanced hydrogen evolution through the formation of active edge sites. Nano Energy. 49, 634-643 (2018); and (3) Xie J, et al. Defect-Rich MoS₂ Ultrathin Nanosheets with Additional Active Edge Sites for Enhanced Electrocatalytic Hydrogen Evolution. Adv. Mater. 25 5807-5813 (2013). Please refer to the 2nd paragraph in page 9 for

corresponding revisions.

(iii). The peak location change in Figure 2(d) cannot be clearly read in its current version. It is suggested to exclusively show only the Zn 2p 3/2 peak in Figure 2(d) because the Zn 2p 3/2 and 2p 1/2 peaks moved along the x axis with the same value. Thus, no need to include two peaks in this figure.

We thank the reviewer for this valuable comment. Figure 2(d) has been adjusted in the revised manuscript to exclusively show only the Zn 2p 3/2 peak and clearly illustrate the peak location change in different ZnS samples (the same figure is shown below for your reference). After highlighting the Zn 2p 2/3 peaks of different ZnS samples, it was found that the Zn 2p 3/2 peaks of pristine ZnS, 0.368-ZnS, and ex0.368-ZnS were located at 1022.48, 1022.26, and 1022.34 eV. This observation indicates that the coordination numbers of Zn in different ZnS samples follow the order: pristine ZnS > ex0.368-ZnS > 0.368-ZnS (please refer to the 2nd paragraph in page 9 for the revised discussions).

Zn 2p patterns of different ZnS samples

(iv). Figure 2(h) is insufficient to show both the formation of structural and atomic defects in acid-etched ZnS. The authors may consider to redraw this schematic figure to serve a better readability.

We thank the reviewer for this insightful suggestion. Figure 2(h) has been revised correspondingly to give a better illustration on both the structural and atomic defects. The revised figure is shown below for your reference.

Diagrammatical illustration of the structural and atomic defects in different ZnS samples

(v). Figure S13 only showed the lattice fringe along different orientations of the ZnS crystal. This did not agree with the discussions in line 223-225. If the exposure probability of different facets was obtained based on their Miller indices? The compatibility between the figure content and the discussions should be further checked.

We thank the reviewer for this valuable reminder. The exposure probability of different facets was actually determined by the classic BFDH model (a model as proposed by Bravais, Freidel, Donnary, and Harker) being implemented in the Morphology Tools of Materials Studio 2020, which generally supports an inverse correlation between the lattice fringes between the $\langle hkl \rangle$ planes and the exposure ratios of corresponding facets. After the Morphology Tools integrated the growth speeds of different ZnS facets, the (111) surface was found to be the most exposed one with the theoretical exposure ratio

reaching as high as ~79%. Thus, by co-considering the HRTEM observations and the morphological evolution laws of ZnS, ZnS(111) surface was finally taken to conduct the DFT calculations. We have specified the methods used to determine the ZnS slab model for DFT calculations (please refer to the 1st paragraph in page S6 of the supporting information) and revised corresponding discussions in the main texts (please refer to the 2nd paragraph in page 12 and the 1st paragraph in page 13).

(vi). The STEM-HAADF that visualized the edge areas of both pristine and acid-etched ZnS should be added in the supporting information. The main goal is to prove that the amorphous layer outside the nanoparticle as shown in Figure 3(f) & (g) formed after the Hg⁰ adsorbed on ZnS.

We thank the reviewer for this constructive suggestion. The STEM-HAADF that visualized the edge areas of fresh pristine ZnS and fresh acid-etched ZnS have been added in the supporting information and denoted as Figure S20 (the same figures are shown below for your reference). As shown, in fresh samples, no matter being etched by acid or not, no amorphous layer was observed on the surface of ZnS nanoparticles, which demonstrates that the amorphous layer was formed after Hg⁰ was adsorbed on the surface of 0.368-ZnS. We have added similar discussions on this observation in the 1st paragraph in page 15.

STEM-HAADF images of pristine and acid-etched ZnS (fresh samples)

(vii). What the numbers in the bracelets in line 316 denotes should be specified.

We thank the reviewer for this meticulous reminder. The numbers in the bracelets in line 316 represent the breakthrough threshold of the designated sorbents. Specifically, the numbers denote the value of normalized Hg^0 concentration $\times 100\%$ at the time when the adsorption capacity of the designated sorbents was calculated. Related explanations have been added in the revised manuscript (please refer to the 1st paragraph in page 16).

(viii). In line 325-327, the inhibitive effects of SO_2 and H_2O were attributed to the competitive adsorption between $\text{SO}_2/\text{H}_2\text{O}$ and Hg^0 . More literature evidences should be provided to support this speculation. Besides, if there is any other reason may explain the inhibitive effects of SO_2 and H_2O ?

We thank the reviewer for these insightful comments. We have added three literature evidences to support that the inhibitive effects of SO_2 and H_2O were attributed to the competitive adsorption between $\text{SO}_2/\text{H}_2\text{O}$ and Hg^0 , i.e., (1) Liao Y, et al. One Step Interface Activation of ZnS Using Cupric Ions for Mercury Recovery from Nonferrous Smelting Flue Gas. *Environ Sci Technol* 53, 4511-4518 (2019); (2) Liu H, et al. Disordered MoS_2 Nanosheets with Widened Interlayer Spacing for Elemental Mercury Adsorption from Nonferrous Smelting Flue Gas. *ACS ES&T Engineering* 1, 1258-1266 (2021); and (3) Xu H, et al. Enhancing the catalytic oxidation of elemental mercury and suppressing sulfur-toxic adsorption sites from SO_2 -containing gas in Mn-SnS_2 . *Journal of Hazardous Materials* 392, 122230 (2020). In addition to competitive adsorption, we also previously found that highly concentrated H_2O might cover the active sites and prevent Hg^0 from contacting with them, thus inhibiting the Hg^0 adsorption performance of metal sulfides (Yang Z, et al. Multiform Sulfur Adsorption Centers and Copper-Terminated Active Sites of Nano-CuS for Efficient Elemental

Mercury Capture from Coal Combustion Flue Gas. *Langmuir* 34, 8739-8749 (2018); Yang Z, et al. Magnetic Rattle-Type Fe₃O₄@CuS Nanoparticles as Recyclable Sorbents for Mercury Capture from Coal Combustion Flue Gas. *ACS Applied Nano Materials* 1, 4726-4736 (2018)). This reason has also been added in the revised manuscript to extend the discussion scope and improve the rigorousness of this work (please refer to the 2nd paragraph in page 16).

(ix). The authors only considered the influence of SO₂ and H₂O on the Hg⁰ capture performances of ZnS. Why the impact of O₂, a flue gas component widely existing in flue gases, was not evaluated?

We thank the reviewer for this practically insightful question. We did not test the impact of O₂ because previous studies have proven that O₂ had negligible influence on the Hg⁰ capture performance of ZnS (Li et al., Effect of Nitrogen Oxides on Elemental Mercury Removal by Nanosized Mineral Sulfide. *Environ. Sci. Technol.* 51, 8530-8536 (2017); Liao et al., One Step Interface Activation of ZnS Using Cupric Ions for Mercury Recovery from Nonferrous Smelting Flue Gas. *Environ. Sci. Technol.* 53, 4511-4518 (2019); and Xie et al., In-situ preparation of zinc sulfide adsorbent using local materials for elemental mercury immobilization and recovery from zinc smelting flue gas. *Chem. Eng. J.* 429, 132115 (2022)). Actually, in addition to ZnS, O₂ was also found to have negligible effect on the Hg⁰ capture performances of all metal sulfides including copper sulfide (CuS) (Yang et al., Multifunctional Sulfur Adsorption Centers and Copper-Terminated Active Sites of Nano-CuS for Efficient Elemental Mercury Capture from Coal Combustion Flue Gas. *Langmuir* 34, 8739-8749 (2018)), molybdenum disulfide (MoS₂) (He et al., Remove elemental mercury from simulated flue gas by flower-like MoS₂ modified with nanoparticles MnO₂. *Chem. Eng. J.* 412, 128588 (2021)), iron sulfides (FeS_x) (Liu et al., Development of Recyclable Iron Sulfide/Selenide Microparticles with High Performance for Elemental Mercury Capture from Smelting Flue Gas over a Wide Temperature Range. *Environ. Sci.*

Technol. 54, 604-612 (2020)), and cobalt sulfides (CoS_x) (Quan et al., Study on the regenerable sulfur-resistant sorbent for mercury removal from nonferrous metal smelting flue gas. Fuel 241, 451-458 (2019)). Considering the extensively-evidenced negligible influence of O_2 on the Hg^0 capture performances of metal sulfides, we thus did not consider to involve this gas component in this work. Related justifications have also been added in the revised manuscript (please refer to the 2nd paragraph in page 16).

(x). The specific experimental conditions for the preparation of MoS_2 should be elaborated. For example, at what temperature this sample was hydrothermally treated, for how long time, and so on.

We thank the reviewer for this constructive comment. The preparation procedures of pristine and acid-etched MoS_2 were specified in the revised supporting information (please refer to the 1st paragraph in page S2). Specifically, for the hydrothermal preparation of pristine or acid-etched MoS_2 , 2.47 g ammonia molybdate was dissolved in deionized water or 0.368 mol L^{-1} H_2SO_4 solution, and then 2.14 g thiourea was added. After stirring for 10 minutes, the mixed solution was transferred into a 100 ml Teflon-lined stainless-steel autoclave and maintained at 220 °C for 18 h. Subsequently, the reaction system was cooled to room temperature, and the precipitate was repeatedly washed by deionized water and ethanol for several times and dried in vacuum at 60 °C for 12 h.

(xi). The Hg^0 breakthrough curves for figure 5(e) should be provided in supporting information.

We thank the reviewer for this constructive suggestion. The Hg^0 breakthrough curves for the abovementioned figure have been added in the revised supporting information and denoted as Figure S22. The same figure is shown below for your reference. Besides, related discussions have also been

revised in the manuscript (please refer to the 2nd paragraph in page 17).

The Hg⁰ breakthrough curves of (a-d) pristine CoS, NiS, CdS, and MoS₂, and (e-h) acid-etched CoS, NiS, CdS, and MoS₂

(xii). Insufficient evidence was provided to prove that CuS possesses the highest Hg⁰ adsorption capacity among all metal sulfides. A comparison table showing the Hg⁰ adsorption capacities of different metal sulfides may be presented in the supporting information.

We thank the reviewer for this valuable suggestion. A comparison table showing the Hg⁰ adsorption capacities of different metal sulfides in their pure phases has been added in the supporting information and denoted Table S1 (the same table is shown below for your reference). As shown, the Hg⁰ adsorption capacity of CuS with the breakthrough threshold equaling to 50% was ~53 mg g⁻¹, higher than all of the other metal sulfides by at least several folds. This table fully justifies that CuS possesses the highest Hg⁰ adsorption capacity among all metal sulfides. Related discussions have been revised in the updated manuscript (please refer to the 2nd paragraph in page 4).

Hg⁰ adsorption capacities of different metal sulfides

Sorbents	breakthrough threshold	Hg ⁰ adsorption capacities (mg g ⁻¹)	References
CuS	50%	52.88	1
ZnS	50%	0.498	2
FeS _x	4%	0.22	3
CoS _x	25%	2.07	4
MoS ₂	90%	16.26	5
PbS	NA	2.76	6
Mn-SnS ₂	NA	0.57	7

1. Yang Z, *et al.* Multiform Sulfur Adsorption Centers and Copper-Terminated Active Sites of Nano-CuS for Efficient Elemental Mercury Capture from Coal Combustion Flue Gas. *Langmuir*. **34**, 8739-8749 (2018).
2. Li H, Zhu L, Wang J, Li L, Shih K. Development of Nano-Sulfide Sorbent for Efficient Removal of Elemental Mercury from Coal Combustion Fuel Gas. *Environ. Sci. Technol.* **50**, 9551-9557 (2016).
3. Liao Y, *et al.* Recyclable Naturally Derived Magnetic Pyrrhotite for Elemental Mercury Recovery from Flue Gas. *Environ. Sci. Technol.* **50**, 10562-10569 (2016).
4. Liu H, *et al.* High-efficient adsorption and removal of elemental mercury from smelting flue gas by cobalt sulfide. *Environ. Sci. Pollut. Res. Int.* **26**, 6735-6744 (2019).
5. Liu H, *et al.* Disordered MoS₂ Nanosheets with Widened Interlayer Spacing for Elemental Mercury Adsorption from Nonferrous Smelting Flue Gas. *ACS ES&T Eng.* **1**, 1258-1266 (2021).
6. Hong Q, Liao Y, Xu H, Huang W, Qu Z, Yan N. Stepwise Ions Incorporation Method for Continuously Activating PbS to Recover Mercury from Hg⁰-Rich Flue Gas. *Environ. Sci. Technol.* **54**, 11594-11601 (2020).
7. Xu H, *et al.* Enhancing the catalytic oxidation of elemental mercury and suppressing sulfur-toxic adsorption sites from SO₂-containing gas in Mn-SnS₂. *J Hazard Mater.* **392**, 122230 (2020).

Reviewer #2

This work proposed a very efficient and meaningful method to overcome one of the major challenges impeding the application of most metal sulfides for Hg⁰ remediation. Amazing outcomes showed that the Hg⁰ adsorption capacities of all metal sulfides pretreated by in-situ etching method reached the level of tens of milligram per gram, approaching or surpassing the critical value defined by CuS. The meaningfulness and contribution of this valuable work indicates its suitability to be published on Nature Communications after some minor revisions are made.

1. The in-situ dissolution of samples as shown by Figure 1(g) is valuable outcomes supporting the boosted Hg⁰ adsorption performance of 0.368-ZnS. Thus, it is suggested to further expand discussions about the influences of the in-situ and ex-situ dissolution process on the textural property of ZnS and its Hg⁰ removal performance.

We thank the reviewer for this great suggestion. The discussions about the influences of the in-situ and ex-situ dissolution processes on the textural property of ZnS and its Hg⁰ removal performance have been expanded in the revised manuscript (please refer to the 3rd paragraph in page 8 and the 1st paragraph in page 9). The same discussion is shown below for your reference.

‘A probable interpretation of this variety is shown in Figure 1(g). During the synthesis of 0.368-ZnS, the nucleation process was accompanied by the *in situ* dissolution of ZnS, which was illustrated by the following equations:

The *in situ* dissolution of ZnS instantly created voids separating the ZnS nanoparticles that were subsequently formed. In this case, as the dissolution of ZnS nanoparticles occurred constantly during nucleation, the undissolved ZnS nanoparticles were efficiently and spatially isolated by the voids, which thus ensured homogenous formation of both structural and surface pores. This favorable homogeneous texture aided the diffusion of heterocomponents into 0.368-ZnS, hence improving the Hg⁰ adsorption performance. However, ex0.368-ZnS was well-crystallized when immersed in acids. In this case, the acid could not efficiently penetrate into the interior parts of the nanoparticles, thus leading to an inhomogeneous distribution in the porous architecture, i.e., only the surfaces of the agglomerated nanoparticles were etched, and no significant changes were observed in their textural properties. The unchanged interior structure containing limited pores might block the transportation of Hg⁰ in ex0.368-ZnS, hence compromising its Hg⁰ capture performance.’

2. The weight loss of ZnS in etching process with 0.368 M is about 50%, now I wonder how about the ZnS yields in other conditions, e.g., 0.092 and 0.184 M. As the author said the Hg⁰ adsorption performance and yield rate of ZnS should be purposefully traded off under practical scenarios. So you need to provide the critical value on this issue.

We thank the reviewer for this insightful comment. The ZnS yields when the acid concentrations equaled to 0.092 M and 0.184 M were 90% and 75%, respectively. Besides, when acid concentration reached approximately 0.75 M, negligible ZnS precipitate was formed in the sample synthesis process. Thus, for ZnS, the critical value of acid concentration was approximately 0.75 M. Related discussions have been added in the revised manuscript (please refer to the 2nd paragraph in page 6 and the 1st paragraph in page 7).

3. The resolutions of XPS deconvolution patterns needs to be improved. The curves colors for Figure 2 (g) needs to be adjusted to make readers easily distinguish their difference. XPS results of sorbents after adsorbing Hg⁰ may be added to further support the interaction between ZnS and Hg⁰.

We thank the reviewer for this valuable suggestion. The resolution of XPS deconvolution patterns have been improved to evidence the valence change of Zn in pristine ZnS and 0.368-ZnS. The curve colors for Figure 2(g) have been adjusted to make readers easily distinguish their difference. The XPS results of sorbents after adsorbing Hg⁰ have also been added and denoted as Figure S14 to further support the interaction between ZnS and Hg⁰. These figures are all shown below for your references. Besides, the discussions on the XPS results of Hg-laden sorbents have been added in the revised manuscript (please refer to the 1st paragraph, page 12). Specifically, the comparison among different

XPS patterns of fresh and Hg-laden ZnS samples not only demonstrated that 0.368-ZnS exhibited superior Hg⁰ accommodation capacity than pristine ZnS, but also suggested the strong interactions between Hg and Zn/S in ZnS. Compared to fresh ZnS, the abundance of under-coordinated sulfur in 0.368-ZnS decreased by 9%, and the Zn 2p peak downshifted by approximately 0.3 eV. This observation suggested that both Zn and S atoms in ZnS played fundamental roles in Hg⁰ adsorption.

Zn 2p patterns of different ZnS samples

Zn/S ratios in different samples

XPS patterns of Hg-laden ZnS samples

4. Could ZnSO₄, a possible product after adding H₂SO₄, be washed out and its role be fully ruled out? Although the XRD results showed no peaks belonging to ZnSO₄, it is possible that the peaks of ZnSO₄ overlapped with the broad peaks of ZnS. In the S 2p XPS image, signals beyond 166 eV were not presented. It is hard to judge if there was ZnSO₄ left in the sample (XPS signal locating at around 168 eV). It is suggested to provide S 2p signal at 168 eV to rule out the influence of ZnSO₄.

We thank the reviewer for this insightful suggestion. The S 2p XPS image showing the signals beyond 166 eV have been added in the revised supporting information and denoted as Figure S8 (the same figure is shown below for your reference). As shown, in the range of 166-169 eV, there was no peak observed for pristine ZnS, 0.368-ZnS, and ex0.368-ZnS, which fully excludes the existence of residual ZnSO₄ in these samples. Related discussions have been added in the 2nd paragraph in page 9.

XPS patterns of S 2p of different samples

5. Can any application implication be derived from the amazing theoretical outcomes of this manuscript? A paragraph discussing the application implication of these outcomes may be added before the conclusion is derived.

We thank the reviewer for this valuable comment. A paragraph discussing the application implication of these outcomes has been added before the conclusion was derived (please refer to the 3rd paragraph in page 17 and the 1st paragraph in page 18). The same discussions are shown below for your reference.

‘Considering the facile procedure and outstanding performance of the *in situ* acid etching method, it may significantly extend the applicability of metal sulfides for heterocomponent adsorption, as mentioned above. Hg^0 is a typical example, for which removal is currently a global concern but lacks flexible and versatile countermeasures. For the effective capture of Hg^0 from industrial flue gases, traditional activated carbons have been gradually replaced by metal sulfides because they contain abundant sulfur ligands, which are Lewis acid, that stably adsorb Lewis bases such as Hg^0 . However, due to the unfavorable coordination environment and pore structure, most of the sulfur ligands in metal sulfides are not available for Hg^0 accommodation. After treatment by the *in situ* acid etching process, the sulfur ligands in most metal sulfides can be effectively activated, which offers more options for abating anthropogenic Hg^0 pollution and recycling the mercury resources. For instance, in removal of Hg^0 from various industrial processes, such as coal combustion, waste incineration, and cement production, the operating temperatures are generally higher than 120 °C, and the Hg^0 removal performance of CuS is largely compromised. With application of the *in situ* acid etching method, sorbents such as ZnS and CdS that exhibit Hg^0 adsorption capacities comparable to that of CuS within the temperature range 120 to 200 °C will be feasible alternatives to activated carbons in the future. In addition, geological access to different metal precursors and

the preferences of different industries can be taken into consideration when choosing the optimal technology for Hg⁰ removal and recovery, considering the mercury loading capacities of in situ etched TMSs approach, or even exceed the loading capacities of the natural sulfide ores (generally ranging between 0.3%-10%) widely used for mercury smelting and production⁴⁸. Finally, the adsorption behaviors of heterocomponents such as NH₃ and CO, which are Lewis bases like Hg⁰, are also expected to be significantly improved by *in situ* acid etching of metal sulfides. This prospect may further extend the opportunities for application of this synthetic method to catalytic energy conversion and storage.'

6. The calculation on the exposure ratios of different crystal surfaces is interesting and meaningful, what is the calculation formula, I recommend the authors providing it or the relevant reference.

We thank the reviewer for this constructive suggestion. The exposure ratios of different crystal surfaces were obtained with the assistance of the Morphology Tools as implemented in Materials Studio 2020 based on classic BFDH model (a model as suggested by Bravais, Freidel, Donnary, and Harker). This model generally supports an inverse correlation between the lattice fringes between the <hkl> planes and the exposure ratios of corresponding facets. From this perspective, the theoretical exposure ratio of the ZnS(111) surface as obtained based on the BFDH model being implemented in the Morphology Tools in Materials Studio 2020 reached as high as ~79%, which is the dominant surface in ZnS crystal and in line with the TEM results. We have specified the methods used to determine the ZnS slab model for DFT calculations (please refer to the 1st paragraph in page S6 of the supporting information) and revised corresponding discussions in the main texts (please refer to the 2nd paragraph in page 12 and the 1st paragraph in page 13).

7. The authors only showed the common analysis results to support the interaction between Hg and ZnS in the DFT calculations. It is good but not enough. Partial density orbital states (PDOS) may be a better method to elucidate this interaction from electronic level. Thus, PDOS results are suggested to be added.

We thank the reviewer for this valuable comment. The electronic analysis based on partial density of state (PDOS) patterns of different adsorption configurations have been added in the revised supporting information and denoted as Figure S19 to support the different interactions between Hg and ZnS in the DFT calculations (the same figure is shown below for your reference). As shown, the interactive intensity between the electrons of the Hg atom and the electrons of different ZnS(111) surfaces follow the order: Zn-defect > S-defect > intact, which is in line with the order of adsorption energy. Related discussions have been added in the 2nd paragraph in page 13.

Partial density of state (PDOS) analysis of the interactions between Hg and (a) intact ZnS(111), (b) S-defect ZnS(111), and (c) Zn-defect(111) surfaces

8. The resolutions of all panels in Figure 4 needs to be improved.

We thank the reviewer for this great reminder. We have revised this figure from one row containing four sub-figures into one row containing two to three sub-figures. Thus, the resolutions of all panels were significantly improved to serve a better readability (the same figure is shown below for your

reference).

(a) influence of reaction temperature on the Hg⁰ removal performance of 0.368-ZnS; the Hg⁰ accommodation capacities of pristine (b) ZnS and (c) 0.368-ZnS (inserted with the real-time Hg⁰ removal efficiency of different sorbents); (d) the Hg⁰ accommodation capacity comparison among different ZnS-based sorbents and nano-sized CuS; the influence of flue gas components on the Hg⁰ capture performances of pristine (e) ZnS and (f) 0.368-ZnS; (g) the effect of acid species on the Hg⁰ capture performance of acid-etched ZnS; and (h) a diagrammatical illustration of the structural motif of acid-etched ZnS

9. What was the percent of under-coordinated sulfur in CuS? Did CuS form hierarchical porous

structure to promote the Hg^0 transportation? In other word, whether the structural properties of CuS were inherited by the acid-etched ZnS to make the assumption in the introduction section reasonable?

We thank the reviewer for this insightful question. The percent of under-coordinated sulfur in CuS generally exceeds 50% (Shen et al., High-Energy Interlayer-Expanded Copper Sulfide Cathode Material in Non-Corrosive Electrolyte for Rechargeable Magnesium Batteries. *Adv. Mater.* 32, 1905524, (2020); Mulla et al., Economical and Facile Route to Produce Gram-Scale and Phase-Selective Copper Sulfides for Thermoelectric Applications. *ACS Sus. Chem. Eng.* 8, 14234-14242 (2020); Yang Z, et al. Multiform Sulfur Adsorption Centers and Copper-Terminated Active Sites of Nano-CuS for Efficient Elemental Mercury Capture from Coal Combustion Flue Gas. *Langmuir* 34, 8739-8749 (2018)). This compositional ratio indicates that, to significantly enhance the Hg^0 adsorption performances of metal sulfides other than CuS, their contents of under-coordinated sulfur must be significantly increased. In this work, the compositional ratio of under-coordinated sulfur in 0.368-ZnS reached 56.8%, approaching that in CuS, which suggests that the *in-situ* acid-etched ZnS adequately inherit the abundant property of under-coordinated sulfur of CuS (as shown in the following figure for your reference).

The S 2p XPS patterns of different ZnS samples

Besides, for the structure of CuS, previous studies well-identify its layered architecture that grows along the <110> planes (Zheng et al., Favorably adjusting the pore characteristics of copper sulfide by template regulation for vapor-phase elemental mercury immobilization. *J. Mater. Chem. A* 10, 10729-10737 (2022); Shalabayev et al., Sulfur-Mediated Mechanochemical Synthesis of Spherical and Needle-Like Copper Sulfide Nanocrystals with Antibacterial Activity. *ACS Sus. Chem. Eng.* 7, 12897-12909 (2019); Yang Z, et al. Multiform Sulfur Adsorption Centers and Copper-Terminated Active Sites of Nano-CuS for Efficient Elemental Mercury Capture from Coal Combustion Flue Gas. *Langmuir* 34, 8739-8749 (2018)). The layered architecture spontaneously separated the CuS nanosheets and preferentially led to the formation of mesopore-enriched structure. Thus, even no morphological control strategy was adopted, the CuS sorbent as prepared possessed an average pore diameter exceeding 10 nm (Liu et al., Recyclable CuS sorbent with large mercury adsorption capacity in the presence of SO₂ from non-ferrous metal smelting flue gas. *Fuel* 235, 847-854 (2019)). In this work, the *in-situ* acid-etching successfully and significantly enlarged the pore size in ZnS (as shown below for your reference). From this perspective, the *in-situ* acid treatment also introduced the structural advantage of CuS into ZnS.

The pore distributions in (a) pristine ZnS, (b) 0.368-ZnS, and (c) ex0.368-ZnS

10. Figure S17 is a fundamental one that explains the roles of structural motif in the Hg⁰ adsorption

by ZnS. Such significance makes this figure more suitable in the main content instead of supporting information.

We thank the reviewer for this meaningful reminder. Figure S17 has been shown in the manuscript and denoted as Figure 5(h) after revision.

11. The authors are recommended to check the consistency of the tense of verbs used in this manuscript.

We thank the reviewer for his/her reminder after carefully checking our manuscript. We have double-checked the consistency of the tense of verbs used in this manuscript and made corresponding revisions. Besides, the manuscript has also been proofread and revised by native English speaking editors at Springer Nature Author Services (SNAS) (please refer to the following figure for the certificate issued by SNAS).

Language editing certificate issued by SNAS

The mercury capture by transition metal sulfides (TMSs) is supposed to be potentially applicable in industrial flue gas cleaning and mercury-laden waste decontamination, acting as potential alternatives to activated-carbon-based techniques considering the cost-effectiveness and eco-friendliness. This paper presents the synthesis of the one-step in-situ acid-etching method. The authors mixed mild acids with TMS throughout the sample preparation process. ZnS was taken as a model material to explore the mechanistic fundamentals responsible for the capacity improvement, evidencing that the increased abundance of surface-active sites and the optimized architecture composed of the porous structure formed a favorable motif that served the overall benefits of Hg⁰ sequestration. Considering the impact and quality, this manuscript seems to be timely and contains important information for academic and industrial researchers. Hence, I support its acceptance by Nature Communications. However, some issues need to be addressed prior to publication.

The authors should discuss more on the critical issues in the field in the introduction section.

We thank the reviewer for this constructive comment. in the revised introduction section, we have clarified the technical advantages of using transition metal sulfides (TMSs) for Hg⁰ pollution abatement, elaborated the critical challenge impeding their extensive applications, quantified the outperformance of CuS in Hg⁰ adsorption, and suggested the key assumptions that inspire the design of an in-situ acid-etching method (please refer to the 2nd paragraph in page 3, and the 1st & 2nd paragraphs in page 4).

Furthermore, detailed structural characterization of samples, such as FFT analysis from the HRTEM images, and ICP-MS data, are highly required to verify the defective crystal structures of as-prepared samples. In addition, supporting evidence from ensemble analysis with atomic-scale

element mappings, EXAFS, and EPR study would provide a undeniable evidence for the presence of the defects.

We thank the reviewer for this insightful suggestion. We have confirmed the existence of both Zn and S defects in 0.368-ZnS based on quantitative evidences as suggested. Specifically, Zn defects in 0.368-ZnS were proven by scanning-transmission electron microscope (equipped with a high-angle annular dark-field detector, STEM-HAADF), simulated elemental mapping, and inductively coupled plasma mass spectrometry (ICP-MS) results, and the S defects were confirmed by X-ray absorption fine structure spectroscopy (XANES) and extended X-ray absorption fine structure (EXAFS) patterns. The STEM-HAADF was not a decisive evidence for the enrichment of S defects because S atoms are not visible in STEM-HAADF due to its low contrast, while EXAFS is not suitable to characterize the Zn defects because it cannot quantify the coordination environment of S as the S atom has no K space. The methods used to conduct these characterizations have been elucidated in the supporting information (please refer to the 1st paragraph in page S3 of the supporting information). The confirmation process of both Zn and S defects in acid-etched ZnS has been discussed in the revised manuscript (please refer to the 2nd paragraph in page 10 and the 1st-2nd paragraph in page 11). The characteristic results and the discussions to support the quantitative confirmation of both Zn and S defects are shown below for your reference.

‘Since the EDS results only provided indicative evidence for the presence of undercoordinated sites in 0.368-ZnS and the species of the undercoordinated sites were unidentified, more atomic analyses were conducted (as shown in Figure 3). The STEM-HAADF (scanning-transmission electron microscopy equipped with a high-angle annular dark-field detector,) images shown in Figure 3(a) and (b) showed the configurations of Zn atoms on the surfaces of the pristine and acid-etched ZnS

because the S atoms were generally invisible in STEM-HAADF images due to their extremely low contrast. As shown, the Zn atoms in acid-etched ZnS exhibited two obvious changes compared to the pristine sample, i.e., (1) the regular hexagonal arrangements of Zn atoms on the ZnS (111) surface became irregular, and (2) the contrast levels of several Zn atoms decreased. The first observation suggested an enrichment in S defects because only the presence of S defects in ZnS caused the displacement of Zn atoms (as shown by the DFT calculation results in Figure 3(c)), while the second observation was much more straightforward in proving the presence of Zn vacancies. The simulated elemental maps based on the STEM-HAADF patterns of pristine and acid-etched ZnS clearly showed dramatically decreasing contrasts for the Zn atoms in acid-etched ZnS, which is probably attributable to the loss of surface Zn atoms (as shown in Figure 3(d)). In this case, the STEM-HAADF image only showed Zn atoms located in the sublayers, which thus made the intensities of the Zn signals decrease. In addition, the inductively coupled plasma–mass spectrometry (ICP–MS) results also proved the nonstoichiometric nature of Zn and S atoms in the acid-etched ZnS (as shown in Figure 3(e)). Specifically, in acid-etched ZnS, the S content was higher by 3% compared to that of the pristine ZnS, and this was accompanied by a corresponding decrease in the Zn content. The excellent agreement between the STEM-HAADF and ICP–MS results further demonstrated the Zn-defect nature of the acid-etched ZnS.

In addition, as mentioned above, the STEM-HAADF pattern for acid-etched ZnS indicated, albeit indecisively, the presence of S defects. To confirm this, the X-ray absorption spectra (XAS) of the Zn atoms in the pristine and acid-etched ZnS are shown in Figure 3(f-h) to characterize their coordination environments. As shown in Figure 3(f), the extended X-ray absorption fine structure (EXAFS) data showing the coordination environments around the Zn atoms indicated Zn-S coordination with radial distances ranging between those of Zn-Zn and Zn-O bonds. From the X-

ray absorption fine structure spectroscopy (XANES) data for the Zn K-edge (as shown in Figure 3(g)), the valences of the Zn atoms in both the pristine and acid-etched ZnS samples were higher than that seen in Zn foil but lower than that seen in ZnO, further confirming the formation of Zn-S bonds. In addition, it should be specially noted that the subfigure in Figure 3(g) highlights the XANES patterns for pristine and acid-etched ZnS, and the valences of the Zn atoms in acid-etched ZnS were lower than those in the pristine samples, which indicated the formation of S defects that decreased the coordination numbers of Zn atoms in the acid-etched ZnS. The corresponding EXAFS R space fitting results quantified the coordination numbers of Zn atoms as 3.9 and 3.6 in pristine and acid-etched ZnS, respectively (as shown in Figure 3(h)). Based on the excellent agreement between the EDS and XAS patterns, it is reasonable to conclude that *in situ* acid etching introduced S defects into the ZnS. Favorable structural motifs comprised of externally introduced pores and defects were thus constructed in 0.368-ZnS to enable adsorption with abundant migration pathways and active sites.'

(a) & (b) STEM-HAADF images of pristine and acid-etched ZnS (fresh samples); (c) geometrical optimization results of intact, S-vacancy, and Zn-vacancy surfaces (pink: S, violet: Zn); (d) simulated elemental mapping of Zn atoms in pristine and acid-etched ZnS; (e) compositional ratios of Zn and S atoms in pristine and acid-etched ZnS as obtained by ICP-MS; (f) EXAFS patterns of pristine and acid-etched ZnS; (g) Zn K-edge XANES patterns of pristine and acid-etched ZnS; and (h) the corresponding EXAFS R space fitting results for pristine and acid-etched ZnS

Besides, the author should carefully check the whole manuscript. There are several grammar mistakes.

We thank the reviewer for this valuable reminder. We have double-checked the grammar used in this manuscript and made corresponding revisions. Besides, the manuscript has also been proofread

and revised by native English speaking editors at Springer Nature Author Services (SNAS) (please refer to the following figure for the certificate issued by SNAS).

Language editing certificate issued by SNAS

Reviewer comments, further round review –

Reviewer #1 (Remarks to the Author):

The authors have seriously responded the questions and comments by the reviewers including me, and addressed a wonderful rebuttals reasonably and rationally. The revised manuscript is also much improved with the solid supporting information. So, I strongly suggest it can be published as is.

Reviewer #2 (Remarks to the Author):

The authors have carefully revised the paper and addressed the reviewers' main concerns. I think this paper can be accepted under current state.

Reviewer #3 (Remarks to the Author):

The work can be published as it is.